# Epidemiological, clinical, and genomic landscape of coccidioidomycosis in northeastern Brazil

Kelsen Dantas Eulálio[1], Daniel R. Kollath [2], Liline Maria Soares Martins[1], Antonio de Deus Filho[1], Maria do Amparo Salmito Cavalcanti[1], Lucas Machado Moreira [3], Bernardo Guerra Tenório[4], Lucas Gomes de Brito Alves[4], Danielle Yamauchi[5], Ligia Vizeu Barrozo [6], George R. Thompson III[7], Mathieu Nacher [8], Jason E. Stajich [9], Gil Benard [10], Eduardo Bagagli[5], Maria Sueli Soares Felipe[11], Bridget M. Barker [2], Luciana Trilles[3] & Marcus de Melo Teixeira [2,4] ✉

Coccidioidomycosis, listed as a priority mycosis by the WHO, is endemic in the United States but often overlooked in Central and South America. Employing a multi-institutional approach, we investigate how disease characteristics, pathogen genetic variation, and environmental factors impact coccidioidomycosis epidemiology and outcomes in South America. We identified 292 cases (1978–2021) and 42 outbreaks in Piauí and Maranhão states, Brazil, the largest series outside the US/Mexico epidemic zone. The male-to-female ratio was 57.4:1 and the most common activity was armadillo hunting (91.1%) 4 to 30 days before symptom onset. Most patients (92.8%) exhibited typical acute pulmonary disease, with cough (93%), fever (90%), and chest pain (77%) as predominant symptoms. The case fatality rate was 8%. Our negative binomial regression model indicates that reduced precipitation levels in the current ($p = 0.015$) and preceding year ($p = 0.001$) predict heightened incidence. Unlike other hotspots, acidic soil characterizes this region. Brazilian strains differ genomically from other *C. posadasii* lineages. Northeastern Brazil presents a distinctive coccidioidomycosis profile, with armadillo hunters facing elevated risks. Low annual rainfall emerges as a key factor in increasing cases. A unique *C. posadasii* lineage in Brazil suggests potential differences in environmental, virulence, and/or pathogenesis traits compared to other *Coccidioides* genotypes.

Coccidioidomycosis is an invasive mycosis included in the WHO list of priority fungal pathogens. It is endemic and notifiable in the USA but is neglected in Latin America[1]. The disease caused by *Coccidioides immitis* is limited to California, Washington, and Northwestern Mexico, whereas *C. posadasii* infections are broadly distributed in the western USA and Latin America[2]. *Coccidioides* spp. are saprotrophic dimorphic fungi phylogenetically placed in the fungal order Onygenales, subphylum Pezizomycotina, phylum Ascomycota[3]. these dimorphic fungi produce infectious arthroconidia in the soil. Inhalation of these particles by mammals, including humans, triggers a morpho-physiological change leading to pathogenic spherules[4]. In endemic areas, the distribution of the fungus in

the soil is focal and heterogeneous, and animal burrows and archeological sites represent the most common environmental reservoirs[5,6]. Coccidioidomycosis is a prominent infectious disease in the western United States, primarily reported in California and Arizona, and is characterized by a noteworthy case fatality ratio. It is estimated that over 500,000 people are naturally infected per year in the US, and increases in cases have been observed in association with severe droughts[7–9]. The majority of infections are asymptomatic or present as mild cases resembling community-acquired pneumonia, while chronic and severe forms are rare, affecting fewer than 5% of symptomatic cases[10].

Coccidioidomycosis is endemic to arid and semi-arid hotspots in Central and South America but with a lower incidence than in North America[1]. Interestingly, 100 years after the first report of the disease in Argentina, fewer than 1000 total cases have been reported in Central and South America[1]. Coccidioidomycosis is not a notifiable disease in Latin America, so the true incidence is unknown. We posit that the lower incidence in Latin America compared to North America may be attributed to factors such as altered virulence traits, underdiagnosis and underreporting, and variations in bioclimatic and demographic conditions[1,2,7]. Northeastern Brazil has been recognized as an endemic area since 1998, but initial case reports date to the 1970s[11–13]. In Brazil, *Coccidioides posadasii* has been isolated from the environment, humans, dogs, armadillos, and bats[13,14]; however, the true coccidioidomycosis burden in Brazil is unknown. The Northeastern states of Piauí, Ceará, Maranhão, Pernambuco, and Bahia, known for their expansive semi-arid landscapes, have reported only a handful of cases with limited epidemiological information[13].

Due to the limited understanding of the disease, we embarked on an extensive evaluation of coccidioidomycosis in Brazil. Our comprehensive, unbiased, and multi-institutional approach aimed to investigate the clinical profile and genetic variations within the pathogen and explore environmental factors that could influence the epidemiology of coccidioidomycosis in Brazil. Therefore, we (1)

described the largest case series study of coccidioidomycosis to date in South America from Piauí and Maranhão states of Northeastern Brazil; (2) integrated geocoded clinical and veterinary cases along with environmental detections of *Coccidioides*, amalgamating these data sets with bioclimatic and demographic factors. This comprehensive approach allowed us to accurately forecast the geographic extent of the disease in Brazil; and (3) inferred the genotypes of *C. posadasii* in Northeastern Brazil using whole-genome analysis. This study will increase awareness of this severe mycosis in South America; and exploring coccidioidomycosis outside California and Arizona deepens our understanding of the species complex disease dynamics.

## Results

### Exploration of demographic and clinical attributes of patients affected by coccidioidomycosis in Northeastern Brazil

We identified 292 patients with proven coccidioidomycosis between 1978 and 2021. Most (n = 270, 92.5%) were from the state of Piauí, with 77 (34.4%) of the 224 municipalities of the state contributing cases. Twenty-two (7.5%) cases were identified in patients referred from the Maranhão state. Reported cases span both caatinga and cerrado biomes, including the semi-arid cerrado-to-caatinga transition, a zone where both biomes meet and intermingle (Fig. 1). Table 1 contains patient clinical and demographic characteristics. Males were predominantly affected (male-to-female ratio: 57.4:1, p < 0.0001). In alignment with the prevalent Afro-Brazilian heritage in the northeastern region of Brazil, a majority of the patients (78.4%) shared this ancestry, while the proportion of individuals of Caucasian descent was considerably lower (20.9%, p = 5.9E−23). Of note, there were only two indigenous patients (0.7%), both from the Guajajara ethnic group in the state of Maranhão.

Armadillo hunting (n = 266, 91.1%) occurring 4−30 days before the onset of coccidioidomycosis-related symptoms was the remarkable practice identified in Northeastern Brazil (Table 1, Supplementary

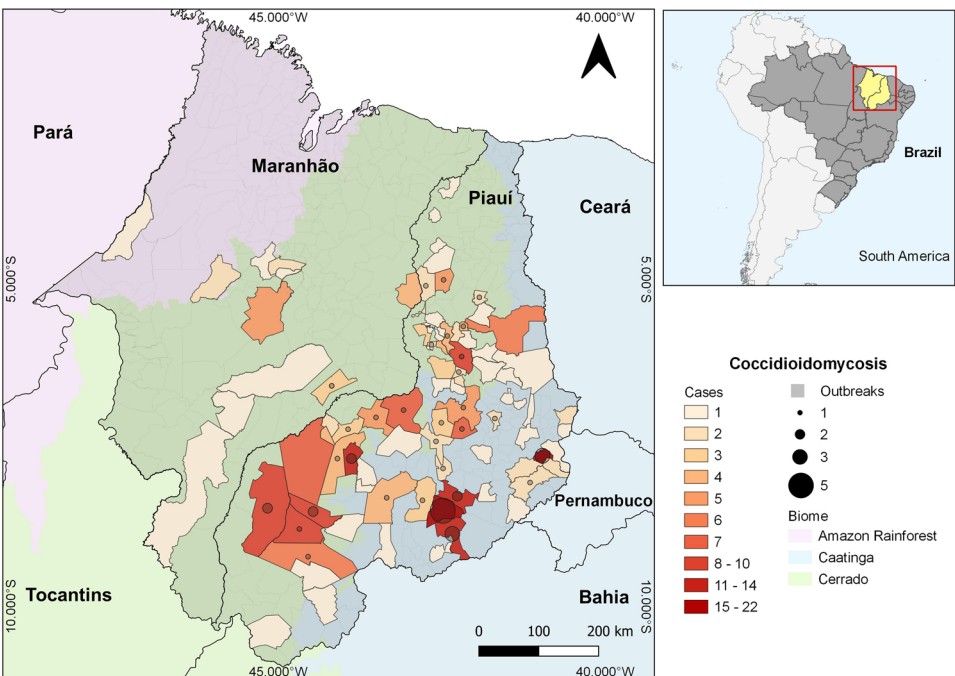

**Fig. 1 | Distribution of coccidioidomycosis cases in Northeastern Brazil.** The 292 coccidioidomycosis cases in Brazil were georeferenced and assigned to each municipality in Piauí and Maranhão states; the scale transitions from pale, the lowest number of cases, to dark red, the highest number of cases. Micro-epidemics are represented by circles within each municipality; the higher the number of

episodes, the greater the size of the circles. The main biomes are also represented by map layers delimited by different color scales: Blue shades represent the Caatinga biome, Green shades represent the Cerrado biome, and purple shades represent the Amazon Rainforest Biome.

**Table 1 | Demographic and clinical attributes of coccidioidomycosis in Brazil**

|  | Number of patients (292) |
|---|---|
| **Demographic characteristics** |  |
| Male sex | 287 |
| Female sex | 5 |
| Age, median years (range) | 31 (7–82) |
| **Origin (state)** |  |
| Piauí | 270 |
| Maranhão | 22 |
| **Ethnicity** |  |
| Afro-Brazilian | 229 |
| Caucasian | 61 |
| Indigenous | 2 |
| **Involvement in potentially high-risk activities** |  |
| Armadillo hunting | 266 |
| Agriculture | 12 |
| Excavation | 6 |
| Gardening | 2 |
| Construction | 1 |
| Fomites | 1 |
| Non identified | 4 |
| **Clinical forms** |  |
| Acute pulmonary | 271 |
| Disseminated | 10 |
| Chronic pulmonary | 7 |
| Pulmonary regressive | 4 |

Fig. 1). Farming was the leading occupation (167/57.2%) followed by students (57/19.5%), trade workers (12/4.1%), and a range of other activities. The most affected age group was 20–39, corresponding to ~50% of the patients; overall, 72.6% were under 40 years old. Coccidioidomycosis occurred as single cases in 63.4% (*n* = 185) of the instances and as microepidemics (two to six patients, average: 2.5– Fig. 1) in 107 instances (39.6%). Microepidemics were related to armadillo hunting of both *Dasypus* sp. and *Euphractus* sp. in 41 of the 42 recorded micro-epidemics; in one case, it was related to stone extraction for civil construction. Finally, of 65 dogs that participated in hunting, 23 developed coccidioidomycosis (35.4%), of whom eight died of the disease.

## Acute pulmonary disease is the main form of coccidioidomycosis in Brazil

Patients' clinical classification was primarily acute pulmonary disease (Table 1; *n* = 271, 92.8%). Ten cases (3.4%) with an initial acute pulmonary disease evolved into disseminated disease. Seven cases (2.4%) presented with chronic pulmonary disease, and 4 cases (1.4%) had residual pulmonary form. More detailed clinical and radiology data were collected from a subset of 100 patients (Table 2). From those, the most frequent symptoms were cough (93%), fever (90%), and chest pain (77%). Cough was predominantly dry (65.6%) or initially dry and increasingly productive in 28 cases (30.1%). Typical community-acquired pneumonia-associated productive cough was rare (4.3%). Dyspnea was reported by 51% of the patients, and 12% had severe pneumonia with acute respiratory failure. Hemoptysis occurred in four cases (4%).

Chest X-rays showed a bilateral multiple nodular pattern in 86.3% of patients. In 52.6%, this was the only radiological observation, in 25.3%, it was associated with parenchymal consolidation, and 7.4% of patients also had interstitial infiltration. Isolated parenchymal consolidation and interstitial infiltration were present in 10.5% and 3.2% of cases, respectively. The nodules were bilaterally distributed, predominating in lung base fields, measuring up to 3 cm in diameter, with ill-defined contours; excavation of the nodules was detected in 10.5% of the cases.

Dermatological manifestations of hypersensitivity were observed. Ten patients presented *erythema nodosum*, 8 *erythema multiforme,* and 6 other types of exanthema; 4 patients had mixed manifestations (Table 2). Mucosal lesions were present in four patients with *erythema multiforme*. Among the 7 patients with disseminated disease, cutaneous/subcutaneous involvement was observed in 3, lymph node involvement in 2, and CNS involvement in 2 cases, one of the latter with concomitant osteo-articular involvement. Azole antifungals were the most used medications alone (55%) or following deoxycholate amphotericin B therapy (24%): itraconazole was preferentially used (71%), followed by ketoconazole (6%), and fluconazole (2%). Deoxycholate amphotericin B alone was the therapeutic alternative used in 12% with severe manifestations. The therapeutic intervention demonstrated efficacy, leading to the resolution of infection in 92% of the patients. However, eight patients evolved towards spontaneous regression. Another eight patients (age range 15–63, 95% CI = 3.5–15.2%) died due to severe pulmonary involvement with respiratory insufficiency (*n* = 6), involvement of CNS (*n* = 1), or both (*n* = 1).

Reassessment for post-treatment sequelae was carried out one to seven years (4.8 ± 2.1) post anti-fungal therapy withdrawal in 25 randomly recruited patients, 24 with prior acute pulmonary disease and one with disseminated disease (cervical lymph node enlargements). Their age range was 13–51 years old. They were all asymptomatic and with no abnormalities on physical examination. However, thoracic imaging revealed residual pulmonary abnormalities in 92% of the evaluated patients. The main abnormalities were multiple nodules (88%), calcified nodules (52%), and fibrosis (16%).

## The role of low annual precipitation in coccidioidomycosis incidence in Northeastern Brazil

All coccidioidomycosis patients, including those involved in micro-epidemics, were from Northeastern Brazil. Beyond the Caatinga biome, which was extensively associated with the disease in Brazil, we have found cases in the Brazilian Savanna biome (Fig. 1). Northeastern Brazil is known for its susceptibility to periods of drought. The region experiences recurrent episodes of prolonged dry spells, which can have a significant impact on local climate conditions and environmental factors[15]. The distribution of cases by diagnosis period varied over time, showing a progressive incidence increase: 16 cases in the 1990s (5.5%), 95 cases (32.5%) in the 2000–2010 period, and 180 cases (61.6%) cases between 2011 and 2021 (Fig. 2A). Since 2010, the lowest number of cases occurred in 2020 (5 cases), potentially due to underreporting during the COVID-19 pandemic. Distribution of cases across time was uneven: in 2004 and between 2015 and 2017, there was a spike in cases, particularly in the state of Piauí (Fig. 2A). We detected a significant inverse association between coccidioidomycosis cases and precipitation in the current year (Fig. 2B, *p* = 0.015) and one-year prior (Fig. 2C, *p* = 0.001). The point estimates calculated by the model for precipitation in the current and in the prior year were both −0.01. Exponentiating those coefficients, we find that, holding temperature and precipitation in the prior year constant, every millimeter increase in precipitation in the current year is associated with 1% fewer coccidioidomycosis cases (IRR: 0.99, 95% CI: 0.98, 0.99). Similarly, holding temperature and precipitation in the current year constant, every millimeter increase in precipitation in the prior year is associated with 1% fewer coccidioidomycosis cases (IRR: 0.99, 95% CI: 0.98, 0.99). Temperature was not a significant predictor (*p* = 0.87), although there does appear to be a slight negative correlation between higher

**Table 2 | Clinical symptoms of patients diagnosed with coccidioidomycosis in Brazil**

| Clinical symptoms | Patients (%) |
|---|---|
| Cough | 93 |
| Fever | 90 |
| Thoracic pain | 77 |
| Adynamia | 59 |
| Anorexia | 53 |
| Dyspnea | 51 |
| Headache | 37 |
| Weight loss | 33 |
| Myalgia | 31 |
| Arthralgia | 28 |
| Erythema nodosum | 10 |
| Polymorphic erythema | 8 |
| Abdominal pain | 7 |
| Cutaneous hash | 6 |
| Oral lesions | 4 |
| Vomiting | 4 |
| Hemoptysis | 4 |
| Skin nodules | 3 |
| Mental confusion | 2 |
| Convulsion | 2 |
| Lymphadenopathy | 2 |
| Arthritis | 1 |

temperatures and cases. The municipalities' altitude ranged from 72 to 600 m above the sea, and the mean temperature ranged from 24.2 to 27.4 °C. Based on data from 1978 to 2021, minimum annual precipitation in Maranhão varies from 961 to 2333 mm, while in Piauí from 640 to 1520 mm. Cases of coccidioidomycosis occurred only below the threshold of 1500 mm, according to the patient's municipality of residence (Fig. 3). In areas with precipitation above 1520 mm, we haven't observed cases of the disease according to both coccidioidomycosis prevalence density and average precipitation maps for the 1978 to 2021 period (Fig. 3).

Species niche modeling suggests that soil characteristics suitable for the fungus exist in areas where coccidioidomycosis cases have not yet been reported, indicating the potential for a larger endemic area than previously observed. According to the logistic regression model output, suitable habitat for *Coccidioides* exists in eastern Maranhão, Piauí, and Ceará states, which represents the known endemic areas of the disease in Brazil and overlaps with our case series. However, the model also identifies areas in Northern Bahia, Sergipe, Alagoas, Pernambuco, Paraiba, and Rio Grande do Norte states as potentially suitable habitats for coccidioidomycosis, suggesting the presence of soil conditions conducive to the fungus, even in regions where cases have not been documented (Fig. 4).

### Assessment of soil physicochemical characteristics in microepidemic areas

Analysis of the physicochemical properties of soil where microepidemics occurred revealed acidic soils, with acidity ranging from medium (five sites/55.6%) to high (four sites/44.4%) (Table 3). The soil texture was sand (44.4%), clay-sand (44.4%), and clay (11.1%). Electrical conductivity, or soil salinity, was very low in the seven soils examined, being classified as "non-saline" soils; electrical conductivity in samples from the municipalities of Caridade and João Costa was not evaluated for technical reasons. The presence of organic matter was low in five soils (55.6%), medium in three (33.3%), and high in only one (11.1%).

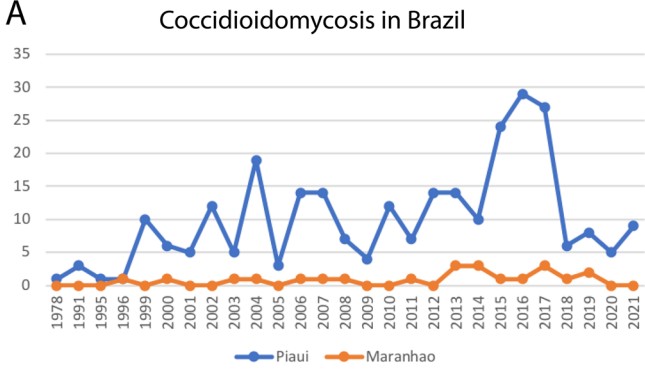

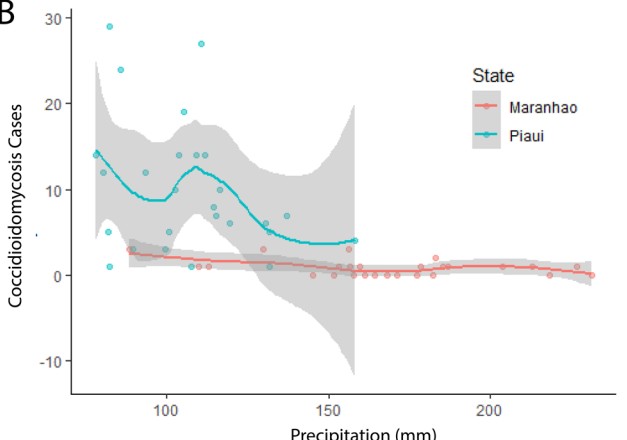

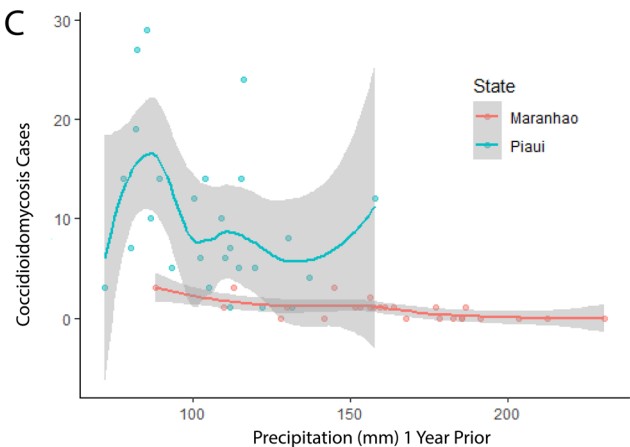

**Fig. 2 | Descriptive epidemiology and climatic predictors of coccidioidomycosis in Northeastern Brazil. A** Time series analysis unveils the evolving distribution of coccidioidomycosis cases in Piauí and Maranhão states over the years. The year intervals are displayed on the *x*-axis, and case counts are displayed on the *y*-axis. Loess curves of precipitation (**B**) and precipitation one year prior (**C**) show a correlation with the increase of coccidioidomycosis cases in Northeastern Brazil. The volume of precipitation is displayed on the *x*-axis in millimeters (mm), and case counts are displayed on the *y*-axis. Error bars represent the 95% confidence intervals.

### Phylogenomic and population genetics reveals a novel *C. posadasii* genotype in northeastern Brazil

The phylogenomic tree obtained recapitulates the genus *Coccidioides* harboring two district species as expected: *C. immitis* and *C. posadasii*. The 13 isolates from Brazil belong to the *C. posadasii* species, more specifically nested within the Texas/Mexico/South America (TX/MX/SA) clade along with the previously published genome from Brazil, B5773[2] (Fig. 5A). Within this clade, all Brazilian

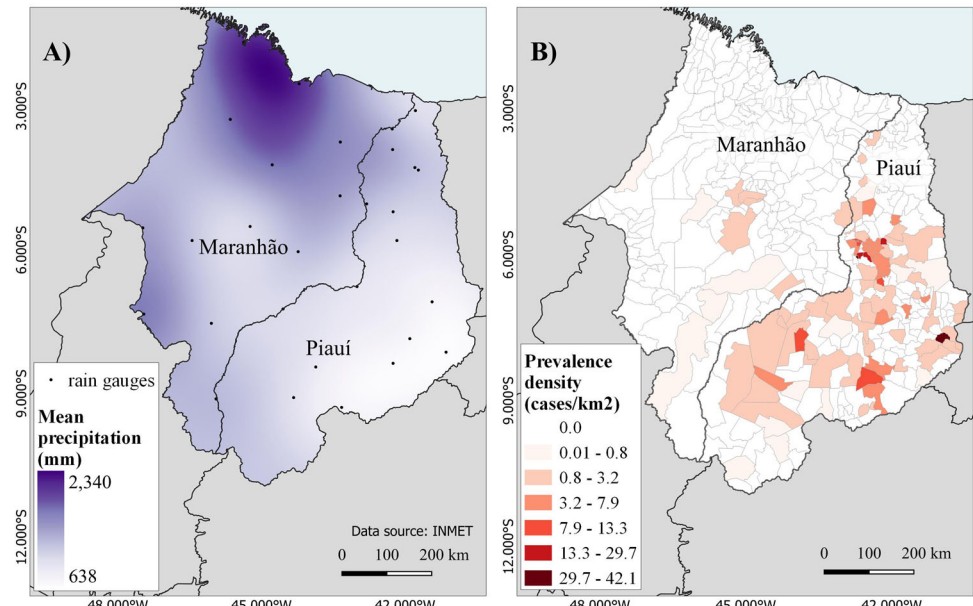

**Fig. 3 | Average precipitation maps and coccidioidomycosis prevalence density in Northeastern Brazil. A** Mean annual precipitation (1978–2021) based on 30 rain gauges in the states of Maranhão and Piauí and **B** prevalence density of coccidioidomycosis by the municipality from the states of Maranhão and Piauí between 1978 and 2021.

strains cluster in a unique branch. Importantly, strains from South America reveal divergent evolutionary trajectories. Strains from Venezuela (CARIBE clade) and Brazil (TX/MX/SA clade) arose from distinct lineages rather than clustering together as a South American clade as deduced by PCA data (Fig. 5B). Low intrinsic genetic variation is inferred from shorter branch length of Brazilian isolates compared to other taxa from the TXMXSA or ARIZONA populations indicating lower number of substitutions per site. The same pattern is observed for the VENEZUELA cluster suggesting multiple and recent introductions in South America. Moreover, the overall nucleotide diversity for Brazilian isolates is low ($\pi = 0.1\%$) compared to the TXMXSA clade ($\pi = 4.5\%$), ARIZONA ($\pi = 5.7\%$) or to GUATE-MALA ($\pi = 1.5\%$) but is similar to that of VENEZUELA ($\pi = 0.09\%$).

Next, we investigated the population distribution of the Brazilian strains compared to other *C. posadasii* individuals using PCA analysis. The PC1 corresponded to most (75.73%) of the total variation and separates *C. posadasii* from *C. immitis*. PC2 corresponded to 4.31% of the total variation and revealed distinct populations within *C. posadasii*. The Brazilian cluster is separated from the ARIZONA, CARIBE and TXMXSA groups, further supporting a unique genetic background compared to other *C. posadasii*.

## Discussion

Coccidioidomycosis was first officially reported in Brazil in the 1970s from Bahia and Piauí[11,12]. However, it was not included in the geographical distribution map until 1998, when acute disease and outbreaks in Piauí and Ceará states were documented[13,16]. Presently, coccidioidomycosis is emerging in Brazil, particularly in Piauí state, where the incidence is increasing[13,17]. This study found that coccidioidomycosis primarily affects young, non-immunocompromised afro-descendent males who acquire the disease by inhaling dust during armadillo hunting, a common practice in the Northeastern states of Brazil (Supplementary Fig. 1). Generally, persons most at risk are from highly marginalized populations and engaged in risky activities. Inhaling a high burden of arthroconidia increases the incidence of severe symptoms[18]. *Coccidioides* spp. are found in animal burrows, where they are thought to degrade organic matter derived from mammals (e.g., keratin, bones, arthropods, excreta) and are well adapted to harsh environments with high mean annual temperature and low water viability[6,19].

This report stands as the most comprehensive case series study of the disease within the South American context. However, a fundamental inquiry persists: What accounts for the disparity in disease prevalence between Latin America and the United States? Coccidioidomycosis is likely underreported due to multiple factors including the lack of (1) availability of rapid, accurate, and reliable diagnostics; (2) reporting to public health authorities; (3) awareness of the disease prevalence—which might be wider than previously known[1,13,20] (Fig. 4). Most cases were diagnosed after significant dust-disturbing activities that produce a high load of dust and arthroconidia, leading to acute pneumonia shortly after exposure. Thus, mild or non-pulmonary forms of the disease might be missed in the current study. Species niche modeling analysis suggest that soils and environmental conditions in other northeastern states of Brazil may be suitable for *C. posadasii* growth and sporulation, and global warming models predict *C. posadasii* will expand[21], so awareness and diagnostic tools may yet find a greater burden of disease.

Even in areas where coccidioidomycosis is more common, frequent misdiagnosis occurs with symptoms erroneously attributed to bacterial and viral pneumonia, tuberculosis, or occupational disease[22]. Thus, there is a significant need to train clinical professionals to "think fungus" and improve diagnostics for coccidioidomycosis at a lower cost. Initiatives like the CDC's "Fungal Disease Awareness Week" campaign are instrumental in cultivating this mycological paradigm shift among professionals. The number of cases reported for other South American countries such as Argentina and Venezuela has long been stably low. Indeed, the low numbers are unlikely to reflect the actual burden due to the lack of compulsory notification and diagnostic tools. Interestingly, cases in Argentina and Venezuela are predominantly classified as chronic forms, either pulmonary or disseminated, and not associated with micro-epidemics[23,24]. In fact, the disease in Brazil resembles North American outbreaks, namely the vast predominance of acute pulmonary disease arising up to 30 days after a high-exposure episode. Signs and symptoms related to acute pulmonary involvement, like fever, dry cough, thoracic pain, dyspnea, and asthenia, among others, overlap with those described in the USA. Also,

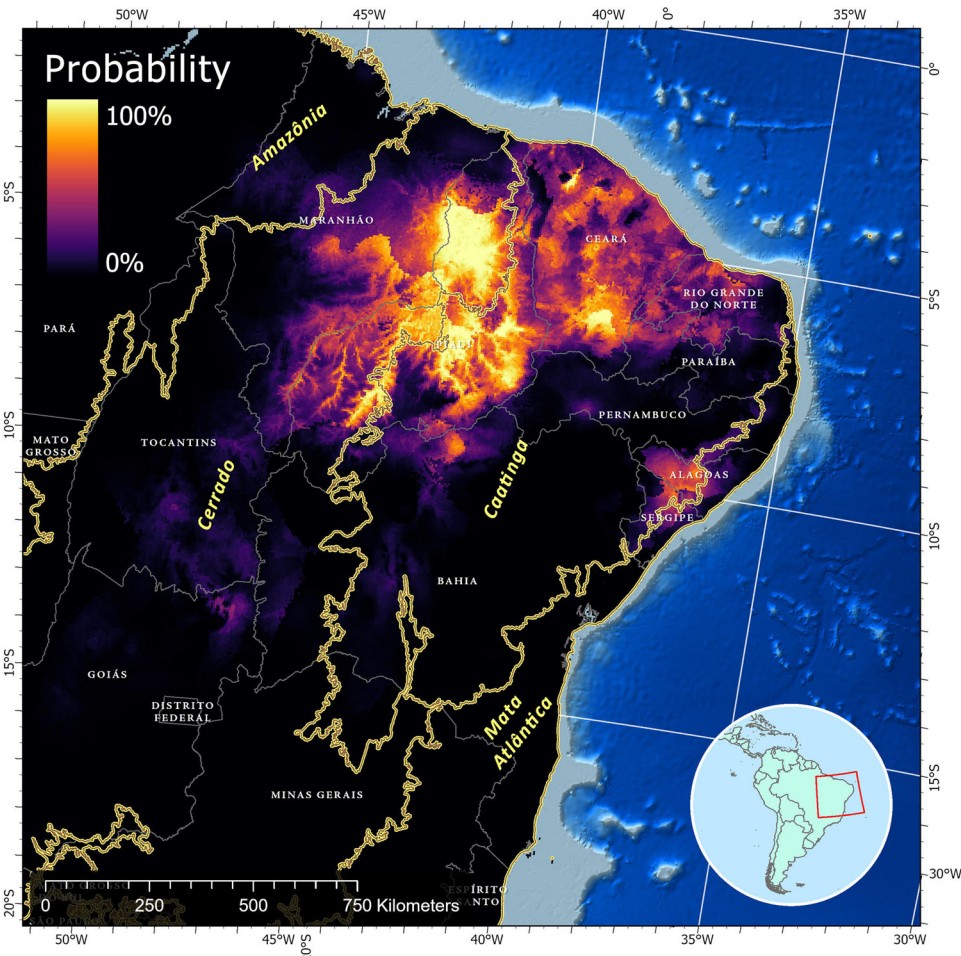

**Fig. 4 | Species distribution modeling of *Coccidioides* sp. in Northeastern Brazil.** Species distribution modeling was conducted using environmental and ecological data to estimate the potential geographic range of *Coccidioides* sp. occurrence. The distribution map is based on a combination of climatic factors, data from field surveys, laboratory analysis, and previous records of *Coccidioides* spp. occurrences and considered in the modeling process. The color gradient on the map represents the probability of *Coccidioides* sp. presence, with lighter shades indicating higher likelihood. The Northeastern states and the main biomes in Brazil are also delimited.

among our 100 cases in whom more detailed data were available, only a subset of patients developed severe disease: 12% with severe pneumonia and acute respiratory failure and 7% with disseminated disease, figures that overlap with those observed for the general population in the southwestern USA, but much lower than those for some specific ethnicities such as African Americans or Filipinos[18,25].

An analysis of predictors of severe disease or mortality, such as the role of ethnicity, could not be carried out due to the high rate of intermixing in the population and because skin color phenotype does not accurately correlate with the patient's genetic background. The resulting fatality rate—8%—was also higher than that reported in other large USA cohorts, e.g., those of Kern County (3%) and Naval Medical Center San Diego (0.9%)[26,27]. The reasons remain to be fully determined and are certainly multifactorial, but presumably include poor access to care and diagnostic delays. Of note, the only two indigenous patients in the cohort had a complicated disease and died; it is essential to highlight that both individuals faced challenges related to medical assistance. Patients in our cohort were treated in accordance with current guidelines, with deoxycholate amphotericin B as the first choice for severe, life-threatening disease and triazoles for the remaining patients. Only a few cases resolved spontaneously. Among azoles, itraconazole was most used, and not fluconazole, perhaps due to Brazilian clinicians' experience in the treatment of other endemic mycoses, e.g., paracoccidioidomycosis and sporotrichosis. The overall

result in our cohort is that treatment was successful, with rare failures related to patients with advanced pulmonary or disseminated disease, similar to other cohorts[28]. Dissemination to CNS was rare.

Thoracic imaging revealed the presence of multiple nodular lesions in 86.3% of the patients, a finding rather uncommon in the USA cohorts[18]. This may reflect exposure to high inoculum. In fact, armadillo burrows represent a rich ecological niche supporting the growth and sporulation of *Coccidioides* spp., which degrade organic matter derived from mammals (e.g., keratin, bones, arthropods, excreta, etc.) and are well adapted to live in harsh environments with high temperatures and low water viability. Armadillo hunting necessarily involves digging deep in the burrows and results in high inoculum exposure. This exposure may relate to the other unexpected finding of persistent alterations in thoracic imaging in almost all (92.1%) patients from a small subset that underwent a post-treatment reassessment. Although clinically symptomatic and having an otherwise normal physical examination, multiple nodules were observed, either calcified or not, which differs from other studies in which residual lesions were lower (2–39.61%)[29,30] and typically presented as single nodules[31].

Patient or environmental factors specific to Piauí and Maranhão do not explain the higher frequency of diagnoses in those states compared to other states in the Northeast. We found the majority of cases in Piauí, in both the Caatinga and the Cerrado biomes. As Caatinga is the dominant biome within the northeast, factors beyond

**Table 3 | Physicochemical properties of soil collected in areas of coccidioidomycosis outbreaks in Northeast Brazil**

| Municipality | Site | pH (acidity) | Organic matter (%) | Texture | Salinity (CE) dS/cm |
|---|---|---|---|---|---|
| Caridade | Armadillo burrow | 5.7 (medium) | 1.11 (low) | Clayey-sandy | Non-determined |
| João Costa | Armadillo burrow | 5.2 (medium) | 2.17 (medium) | Clayey-sandy | Non-determined |
| Sebastião Leal | Armadillo burrow | 4.5 (high) | 0.72 (low) | Sandy | 0.2 (non-saline) |
| Prata do Piauí | Armadillo burrow | 5.7 (medium) | 1.7 (medium) | Sandy | 0.3 (non-saline) |
| Elesbão Veloso | Armadillo burrow | 5.2 (medium) | 0.68 (low) | Clayey-sandy | 0.2 (non-saline) |
| Canto do Buriti | Armadillo burrow | 4.0 (high) | 6.98 (high) | Sandy | 2.5 (non-saline) |
| Monsenhor Gil | Armadillo burrow | 5.2 (medium) | 0.95 (low) | Clayey | 0.3 (non-saline) |
| Altos | Charcoal factory | 4.8 (high) | 1.66 (medium) | Sandy | 0.2 (non-saline) |
| Teresina | Quarry | 4.9 (high) | 1.02 (low) | Sandy | 0.1 (non-saline) |

The location, site, pH, percentage of organic matter, texture and salinity are displayed.

the biome alone must explain lower diagnoses in cases in other northeastern states. We note that both biomes share physiochemical characteristics such as high temperatures and low water viability. Also, soil management for agricultural activities has reduced carbon content and microbial activity, which, combined with mean annual precipitation below 1500 mm, may favor fungal development. The increase in cases in Brazil was associated with low annual precipitation, as indicated by the bioclimatic variables of precipitation and precipitation one year prior (Figs. 2 and 3). It is worth noting that a similar trend has been observed in the United States[32]. In addition to the absence of dust storms and earthquake-induced landslides in Brazil, which are occasionally associated with increased coccidioidomycosis in North America, we have noted important unique physicochemical features of the Northeastern states of Brazil. The soils sampled in areas of outbreaks are acidic and non-saline, contrary to observations in endemic areas of Arizona and California, which are alkaline and saline[33]. Another relevant differential characteristic of the disease between North and South America is the pathogen's genetic background. The disease in Arizona is caused by *C. posadasii* ARIZONA population, while in Central America and Venezuela is caused by *C. posadasii* CARIBE population[2]. The isolates from Brazil form a unique population within *C. posadasii* TXMXSA cluster (Fig. 5). Such genetic differences might reflect specific virulence and pathogenesis traits in North America (caused by *C. posadasii* ARIZONA) relative to South America (caused by *C. posadasii* CARIBE and TXMXSA). Although coccidioidomycosis is not a notifiable disease in Texas or Mexico, the incidence of the disease appears lower than in Arizona and California in spite of similar skin test positivity[34]. It is worth noting that the *C. posadasii* TXMXSA population share a common ancestor, which might contribute to a set of traits that affect the ability to cause disease after infection. Whether or not different *C. posadasii* and *C. immitis* populations display altered clinical traits that are unknown.

While our study provides valuable insights into the epidemiology and characteristics of coccidioidomycosis in Brazil, several limitations should be acknowledged. Firstly, our data are based on a retrospective analysis of cases spanning over four decades, and the accuracy and completeness of historical records may vary. Additionally, our study focused on two states in Northeastern Brazil, which may not fully represent the diversity of coccidioidomycosis across the semi-arid portion of the country, for example, Ceará state and other areas potentially identified as endemic by species niche modeling. Another limitation of our study is that the observed cases may likely be biased towards more severe pulmonary manifestations, as testing for coccidioidomycosis is not mandatory in cases of community-acquired pneumonia, leading to an underrepresentation of milder or asymptomatic infections associated with low arthroconidia exposure as in other case- series[13]. As a result, our findings may not capture the full spectrum of disease presentation and risk

factors. Despite these limitations, our study highlights the significance of coccidioidomycosis in this region and provides a foundation for future research in the field.

Taken altogether, we conclude that coccidioidomycosis in Northeastern Brazil has a specific infection profile associated with armadillo hunting. In Brazil, coccidioidomycosis predominantly impacts marginalized populations, specifically individuals involved in activities like armadillo hunting. This practice holds a unique environmental, economic, and cultural importance, particularly for the subsistence of rural communities in this context. Contrary to North America, where infection risk extends to a diverse demographic, these findings underscore disparities in infection mechanisms and the makeup of affected populations in the two regions. Our observations indicate that particular bioclimatic variables, such as reduced precipitation, may play a pivotal role in the escalation of cases in Brazil. These findings are particularly relevant within the framework of global climate change in the Northeastern portion of Brazil[21]. We also show that a unique *C. posadasii* genotype is circulating in Brazil and we propose that it might display different virulence and pathogenesis traits than other *Coccidioides* genotypes. We suggest that global and federal public and occupational health authorities should support and collaborate with established local centers in these endemic regions to better understand coccidioidomycosis in South America.

## Methods
### Clinical data
This study was approved by the Research Ethics Committee of the Institute of Tropical Diseases Nathan Portela and subsequently ratified by the Ethics Committee of the Federal University of Piauí (No. 36/08-CAAE 0036.0.045.000-8). We conducted a retrospective study of microbiologically proven coccidioidomycosis cases between 1978 to 2021 at the Institute of Tropical Diseases Nathan Portela and at the Pulmonology Clinic of Hospital Getúlio Vargas. In this study, the patients were referred from Maranhão and Piauí states, renowned for their expansive Caatinga and Cerrado biomes. The Caatinga is distinguished by its xerophytic flora, adapted to arid conditions, while the Cerrado biome is recognized for its ecological diversity within the Brazilian savanna, featuring heterogeneous landscapes and abundant biodiversity. Patients showing clinical respiratory signs were evaluated, and inclusion criteria were based on the isolation or observation of fungal cells compatible with *Coccidioides* (Supplementary Methods). None of the patients had a history of immunodeficiency, recurrent infections, or immunosuppressive regimens. We also ruled out tuberculosis in patients suspected of having it. Lung damage was assessed using chest X-ray and/or computed tomography (CT). Residual lesions such as nodules or cavitation and sequelae such as calcifications, fibrosis, or bronchiectasis were investigated. Demographic data and risk activities were recorded (Supplementary Methods).

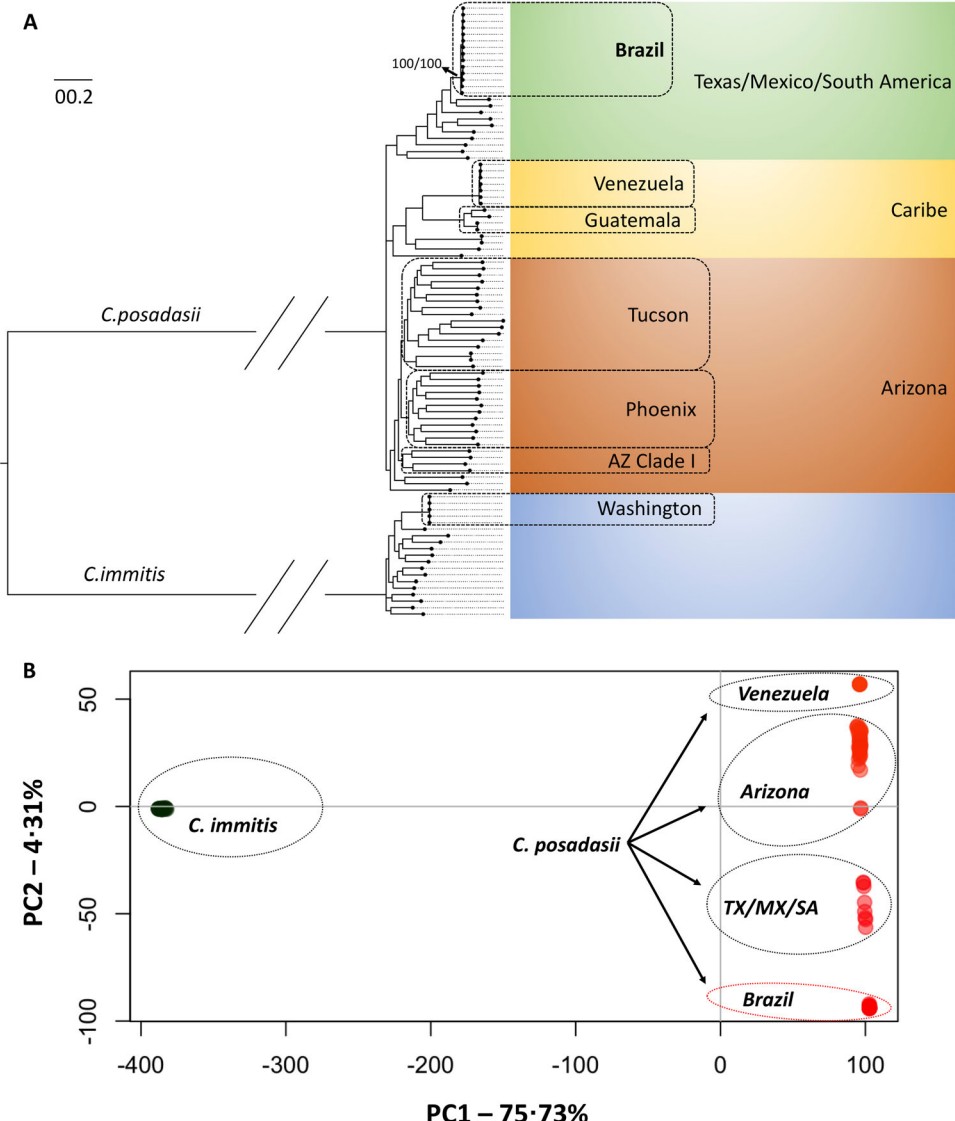

**Fig. 5 | Evolutionary analysis of *Coccidioides posadasii* isolated in Northeastern Brazil. A** The Maximum-likelihood phylogenomic tree shows that clinical isolates from Brazil are nested within *C. posadasii* species in a unique cluster with the Texas/Mexico/South America population. The branches are proportional to the number of mutations and 1000 ultrafast bootstraps and SH-aLRT were used to test branch support and added to the Brazil clade. The nodes in the tree represent common ancestors, and the branching points indicate the divergence of lineages. The tree was rooted with *C. immitis*. **B** Principal coordinate analysis (PCA) shows the genetic distribution of *Coccidioides*, a fungal pathogen. These coordinates are derived from a matrix of whole-genome polymorphisms or dissimilarities between the isolates from diverse geographical locations The PCA plot depicts the distribution of the isolates along the first two principal coordinates; PC1 axis indicates the degree of genetic dissimilarity or similarity between *C. immitis* and *C. posadasii* while PC2 captures a smaller proportion of the total variation of *C. posadasii* clusters.

Hunting armadillos was identified as a common activity prior to infection. Accordingly, for cases with reported exposure to dust from armadillo habitats, we also followed up and confirmed the infection status of any dog involved in the hunting (Supplementary Methods). The cases were classified as microepidemics when two or more human cases were diagnosed with coccidioidomycosis signs and symptoms that appeared simultaneously after a shared exposure event. Shared exposure events were recognized through the occurrence of coccidioidomycosis clinical symptoms in individuals who had been exposed to common infectious sources (i.e., hunting an armadillo in a burrow complex or being involved in extracting stones from a common excavation site). Clinical signs, symptoms, and antifungal treatment of 100 patients with laboratory-proven coccidioidomycosis were retrieved through a review of medical records and prospectively through subsequent clinical reviews. These 100 patients were chosen on the basis of having complete medical records. Clinical symptoms,

including fever, cough, rash, anorexia, chest pain, dyspnea, weight loss, hemoptysis, myalgia, headache, arthralgia, erythema nodosum, erythema multiforme, skin rash, among others, were collected as part of the clinical data. We recorded symptom onset dates and calculated the duration of the disease from symptom onset to confirmatory mycosis diagnosis. Co-morbidities, such as HIV/AIDS, diabetes mellitus, tuberculosis, leukemias, lymphomas, neoplasms, collagen diseases, smoking, and alcoholism, were also examined.

## Niche modeling and assessment of climate variables' influence on coccidioidomycosis in Brazil
We collected patient addresses from our cohort to geocode the entries using QGIS 3.10.9 software (http://www.qgis.org/) with SIRGAS 2000 Datum and a cartographic base from the Brazilian Institute of Geography and Statistics (IBGE 2020). Prevalence density was calculated by collecting the number of cases by municipality and dividing by the

area of the municipality (in km²) multiplied by 1000. We initiated our analysis of niche modeling using 19 bioclimatic variables sourced from the WorldClim database (www.worldclim.org/current). To ensure model validity, we rigorously tested for autocorrelation and selected variables with an autocorrelation coefficient (*r*) below 0.809. Subsequently, we addressed collinearity by examining pairwise Spearman's correlations, followed by a leave-one-out Jackknife test to select the least impacting variable. Our final model, utilizing the Maximum Entropy (MaxEnt) machine learning model, exhibited an AUC of 0·87 after 1000 maximum iterations. We used a negative-binomial regression test to effectively discriminate between sites with and without *Coccidioides* (Supplementary Methods).

In the Northeastern Brazilian states of Maranhao and Piauí, we investigated the relationship between climate and coccidioidomycosis cases from 1978 to 2020 using negative-binomial regression models. Coccidioidomycosis cases were aggregated yearly and at the state level between 1978 and 2020 in both the Maranhao and Piaui states of Northeast Brazil for regression modeling. The model included fixed effects for total yearly precipitation, average yearly temperature, total precipitation from the previous year, and state. The final model is presented in the Supplementary Methods. The model was screened for autocorrelation among predictor variables, ensuring that all retained variables maintained Pearson's correlation coefficients below 0.6. Rainfall data were collected from the public repository of the Instituto Nacional de Meteorologia (https://portal.inmet.gov.br/). Daily rainfall data of the States of Maranhão 187 and Piauí were collected by 30 conventional rain gauges for the period of 1978 to 2021. Annual precipitation (in mm) was obtained by summing the daily data by rain gauge. Data interpolation was made based on the analysis of variograms and ordinary kriging for the average of the period. Calculations and modeling were performed with the R packages 'sf'[35], 'spatstat'[36], 'stars'[37], and 'MASS'[38].

Lastly, we assessed the characteristics of the soil from locations of known exposures. Nine soil samples were gathered from locations where patients reported significant dust exposure prior to exhibiting clinical symptoms, including environments such as armadillo burrows, quarries, and charcoal factories. These samples were then subjected to comprehensive analysis, measuring key physicochemical attributes such as texture, salinity, pH, and total organic content, following established protocols[39] (Supplementary Methods).

### Fungal isolation, genome sequencing, and whole-genome genotyping

To unravel the genetic diversity and population structure of *Coccidioides posadasii* in Brazil, our study conducted fungal isolation, whole-genome sequencing, and evolutionary comparisons with additional *Coccidioides* sp. genomes. Fungal growth was achieved by plating clinical specimens (Supplementary Methods) into the fungibiotic medium Mycosel. Whole genome typing was performed in 13 isolates. DNA was extracted from 500 mg of cells, assessed for integrity via agarose gel electrophoresis, 1 µg of input DNA subjected to library preparation using the NEBNext® Ultra™ II DNA Library Prep Kit (New England Biolabs) and quantified via qPCR, Bionalyzer (Agilent) and Qubit (Invitrogen). DNA libraries were sequenced using the NovaSeq 6000 instrument (Illumina-kit v1.5−300 cycles−2X150bp) in a high-throughput mode. Sequences were deposited in the SRA database under the following accession numbers: SRR25495556-SRR25495568. For evolutionary comparisons, an additional 81 *Coccidioides* sp. genomes were retrieved from previous studies[2,40] (Supplementary Methods). SNPs were retrieved after a series of in silico genome alignments and filtering steps, phylogenomic analysis was performed under maximum-likelihood criteria, and principal coordinate analysis (PCA) was performed to verify the population distribution within *C. posadasii*. Nucleotide diversity was calculated within each *C. posadasii* population (Supplementary Methods).

### Reporting summary

Further information on research design is available in the Nature Portfolio Reporting Summary linked to this article.

### Data availability

The clinical data that support the findings of this study are available from the corresponding author upon reasonable request. The genomic data generated in this study have been deposited in the Bioproject, Biosample, and Sequenced Read Archive (SRA) databases under accession code PRJNA1000610, SAMN36772057-SAMN36772069, and SRR25495556-SRR25495568, respectively.

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

## Acknowledgements

We dedicate this work to the late Dr. Bodo Wanke, whose visionary leadership and extensive contributions to the study of coccidioidomycosis in the Brazilian semi-arid and endemic regions have left an indelible mark. Dr. Wanke's dedication to training individuals, conducting field research, and closely investigating patients has significantly enriched our understanding of this disease. His legacy continues to inspire and guide our work. His passion for research and commitment to advancing medical knowledge will be remembered and honored. We are thankful to the National Council for Scientific and Technological Development (CNPq) for supporting this work under grant number 434640/2018-2 (M.M.T.). B.M.B. is funded by the Arizona Board of Regent's Grant "Understanding Valley Fever and Reducing Dust in Arizona", 5U19AI166798 and 5U19AI166058. G.B. and M.S.S.F own hold senior researcher fellowships from CNPq. G.R.T. is supported by NIH grant number 5U19AI166798-02. J.E.S. is a Canadian Institute For Advanced Research Fellow in the program Fungal Kingdom: Threats and Opportunities and was supported by NIH grants R01 AI127548 and AI130128, the University of California Multicampus Research Programs and Initiatives grants MRP-17-454959 "UC Valley Fever Research Initiative" and VFR-19-633952 "Investigating fundamental gaps in Valley Fever knowledge" and United States Department of Agriculture—National Institute of Food and Agriculture Hatch Project CA-R-PPA-5062-H.

## Author contributions

Kelsen Dantas Eulálio—Conceptualization, data curation, formal analysis, supervision, investigation, methodology, writing—original draft. Daniel R. Kolath—Data curation, formal analysis, visualization. Liline Maria Soares Martins—Data curation, Antonio de Deus Filho—Investigation. Maria do Amparo Salmito Cavalcanti—Conceptualization, investigation. Lucas Machado Moreira—Methodology. Bernardo Guerra—Data curation, formal analysis. Lucas Gomes de Brito Alves—Data curation, formal analysis. Daniele Yamauchi— Data curation, Formal Analysis, Visualization. Ligia Vizeu Barrozo—Data curation, formal analysis, visualization, writing—review & editing. George R. Thompson III—Validation, writing—review & editing. Mathieu Nacher—Validation, writing—review & editing. Jason W. Stajich—Resources, software, methodology. Gil Bernard—Data curation, validation, writing—review & editing. Eduardo Bagagli—Conceptualization, resources, validation. Maria Sueli Soares Felipe—Conceptualization, resources, funding acquisition, writing— review & editing. Bridget Barker—Resources, software, methodology, writing—review & editing. Luciana Trilles —Conceptualization, funding acquisition, investigation, methodology. Marcus de Melo Teixeira— Conceptualization, data curation, formal analysis, supervision, investigation, methodology, writing—original draft, writing—review & editing.

## Competing interests

The authors declare no competing interests.

## Additional information

[1]Hospital de Doenças Infecto Contagiosas-HDIC, Federal University of Piauí-UFPI, Teresina, Piauí, Brazil. [2]The Pathogen and Microbiome Institute, Northern Arizona University, Flagstaff, AZ, USA. [3]Evandro Chagas National Institute of Infectology, Fiocruz - RJ, Rio de Janeiro, Brazil. [4]Faculty of Medicine, University of Brasília, Brasília, Federal District, Brazil. [5]Departamento de Microbiologia e Imunologia, Instituto de Biociências de Botucatu, Universidade Estadual Paulista/UNESP, Botucatu, Brazil. [6]Department of Geography, Faculty of Philosophy, Languages and Literature, and Human Sciences, University of São Paulo, São Paulo, Brazil. [7]Department of Internal Medicine, Division of Infectious Diseases and Department of Medical Microbiology and Immunology, UC-Davis, Sacramento, CA, USA. [8]Centre d'Investigations Cliniques, INSERM 1424, Centre hospitalier de Cayenne – French Guiana, Cayenne, French Guiana. [9]Department of Microbiology and Plant Pathology University of California-Riverside, Riverside, CA, USA. [10]Laboratório de Micologia Medica, Departamento de Dermatologia, Instituto de Medicina Tropical, Faculdade de Medicina, University of São Paulo, São Paulo, Brazil. [11]Universidade Católica de Brasília, Brasília, Brazil. ✉e-mail: marcus.teixeira@gmail.com

