## [Peer Review File · Nature Communications]

Epidemiological, Clinical, and Genomic Landscape of Coccidioidomycosis in Northeastern BrazilReviewers' Comments:

Reviewer #1:

Remarks to the Author:

A very exciting and well-done study. This work addresses a major gap of understanding in coccidioidomycosis in South America/ Brazil. The work clearly reports risk factors, clinical presentations, and genomic epidemiology of the disease in northeastern Brazil. Relatively minor comments below.

Abstract:

Line 56: cases to case

Line 63: spike to increase

Lines 64 and 65: Unclear what the p values represent

Background:

Lines 79 and 80: Indicate what Pezizomycotina and Ascomycota mean for the reader.

Lines 85: Unclear what this sentence is trying to say.

Line 108: Sentence is a fragment.

Methods:

Line 131: Rather than microepidemics, why not outbreaks?

Line 145: Missing period

-Would be good to describe the population seen at the Nathan Portela institute, mostly thinking about geography for patients serviced.

-Detail veterinary (dog) specifics in the clinical data.

-Methods do not indicate where soil acidity data was pulled from or if measured, how. Same with soil text, salinity, and other parameters.

-Where the sequencing data is publicly available is not described.

Results:

Line 165: Be consistent about capitalizing Northeastern

Line 179: Consistency with types of nouns. Farmers and students; suggest saying majority were farmers (X) etc.

Line 181: Indicate years

Line 219: Indicate years

Discussion:

-Where is the data showing that most cases were immunocompetent? If no data, explain why it is assumed.

-Limitations of the study need to be clearly stated including that the sample set is not derived from systematic surveillance of northeastern Brazil, thus, findings may not be representative of coccidioidomycosis in northeastern Brazil.

Major gaps are an unclear description of where the data for soil characteristics comes from or if generated, how it was measured; limitations of the study of which there are key ones given the collection of samples/cases; and submission of genomic data to public repositories.

Reviewer #2:

Remarks to the Author:

This paper examines coccidioidomycosis in Brazil, describing characteristics of patients, genetic variation in clinical isolates, and environmental factors associated with case and environmental detection. Coccidioidomycosis is an emerging infection and of importance to study. A strength of this

article is that it is the largest case series study of the disease in South America. The authors cover a wide range of research questions. I do have some concerns with the conclusions drawn from the study, which are documented under Major comments (and some under minor comments). Additionally, I believe there is opportunity to strengthen this paper by improving clarity in key places and expanding upon methods. In the attached, I document major and minor concerns.

This paper examines coccidioidomycosis in Brazil, describing characteristics of patients, genetic variation in clinical isolates, and environmental factors associated with case and environmental detection. Coccidioidomycosis is an emerging infection and of importance to study. A strength of this article is that it is the largest case series study of the disease in South America. The authors cover a wide range of research questions. I do have some concerns with the conclusions drawn from the study, which are documented under Major comments (and some under minor comments). Additionally, I believe there is opportunity to strengthen this paper by improving clarity in key places and expanding upon methods. Below I document major and minor concerns.

Major comments:

- While it may be a logical assumption to think that droughts are related to coccidioidomycosis in Brazil, I do not believe the findings of this paper can tell us about drought in Brazil for a few reasons, including 1) drought was not modelling in this analysis – only precipitation; 2) I have concerns about model specification, which are documented in the methods section. I suggest that statements about drought being key for coccidioidomycosis in Brazil be replaced with statements only about precipitation. Results should state effect estimates and 95% CI's. If the authors wish to discuss drought, they may do so, but should include citations about drought frequency, periods classified as droughts according to some definition, and climate projections for whether Brazil is expected to become drier or wetter under climate change. Alternatively, I do not believe the paper would suffer from a more descriptive approach to the time series data. For instance, cases were really high in 2015-2017 – what was happening then?
- I worked hard to keep the states and biomes straight. I suggest re-writing several areas of this paper assuming the reader has no background on Brazil geography. I include below some areas where this could be done.
- In some areas, methods are lacking, including, as mentioned, regression model specification, as well as the soil analysis
- The authors cover so many different topics in this paper, which is both a strength and can make the paper difficult to follow. I suggest more clear delineations between research questions.

Minor comments

Abstract:

- Precipitation: suggest to clarify that low precipitation is associated with incidence
- WHO's priority list [of fungal pathogens]
- Line 56: change semi colon to comma
- Line 64: clarify if precipitation is current year precipitation, also clarify the relationship is an inverse one
- Line 66: potentially a typo - *C. posadasii* strains
- Line 68: suggest to use a word other than pluviosity (rainfall?) that might be more recognizable; see comments about "drought"

Background:

- Line 84: would it make more sense to change animal "dens" to "burrows"?

- Line 85: The phrasing of this sentence could be edited for clarity: “The number of infections in California and Arizona ranks disease burden as one of the highest reported infectious diseases, with high morbidity and mortality”. Is this saying, “Coccidioidomycosis is one of the highest reported infectious diseases [nationally? Globally? In CA and AZ?], with most of the infectious reported from California and Arizona, and is associated with high morbidity and mortality.” Secondly, the phrase “associated with high morbidity and mortality” is vague. I might suggest to remove and include the case fatality ratio.
- Line 88: in addition to sources 7 and 8, more recent study has found associations between drought and incidence (Head, et al., *Lancet Planetary Health*).
- Line 96: reported in Argentina?
- Line 108: This sentence seems like a partial sentence, “Our unbiased and multi-institutional approach to assess genetic variation in the pathogen and environmental components that might affect the epidemiology of coccidioidomycosis”
- Line 111: I am a little confused about whether data is from only these two states or all of Brazil, given that results (line 168 and elsewhere) talks about other states.

Methods:

- Line 130: an outsider may not understand dogs are involved in hunting. Maybe edit “the dogs” to “of hunting dogs”
- Line 132: how was a shared event identified?
- Clinical data: how were the 100 patients chosen?
- In section 2, suggest to more clearly delineate your aims here as 1) niche modeling and 2) assess the relationship between climate variables and cases.
- Section 2 Appendix could provide more details about the models. I also suggest at least two separate paragraphs to delineate the two different aims. If you have room. I would suggest to include some details on environmental variables in the main text.
 - o Niche modeling: what were the 19 variables obtained as predictors?
 - o Nice modeling: if two variables were autocorrelated, were both discarded? Or how did you select which ones?
 - o Niche modeling: might be helpful to include the months of the coldest quarter and the warmest quarter. Is precipitation seasonal in Brazil? What does a term on temp seasonality mean? Also clarify that precipitation is total.
 - o Regression model: how did you aggregate case counts (e.g., over what space and time period?)
 - o Regression model: precipitation and temperature in the current year seems troublesome as a variable because, if I am understanding correctly, some of the precipitation/temp factoring into the metric could have occurred following the case. Is it possible to examine finer temporal scales?
 - o Regression model: precipitation over time is often correlated. Is both precipitation and precipitation one year prior put into the model? I might suggest to do so.
 - o Regression model: I imagine over time that population changed. How was population accounted for in your models?
 - o Regression model: what model was selected as the final model? (e.g., what were its components?)

- Section 3: Include sample size for clinical isolates in main text.
- In results you talk about soil characteristics from microepidemics, but this is not well-described in the methods. Can the authors add more about their sampling plan (how many samples were collected and how) and how they tested for various properties (e.g., texture, salinity, pH, organics). Were these soils tested for *C. posadasii*?

Results:

- Line 170: the reader needs more context for caatinga and cerrado biomes. Perhaps a description could be added to introduction, or methods. Similarly, it is not clear to me what a cerrado-to-caatinga transition means. This is also true to Savana (line 233)
- Identification of risk factors: it is difficult to say if these characteristics are risk factors without an understanding of the distribution of these factors in the underlying population.
 - o is it possible to state the demographics of the underlying population? Are roughly 78% of the underlying population African American? (Is African American correct here?)
 - o Similarly, what proportion of the underlying population are farmers? In various age groups?
 - o If this information is not known, I would just describe this as descriptive, rather than “risk factors”
- Line 179: “this area of Brazil” – without familiarity with Brazil, and my confusion about what states this study covers, I am not sure what part of Brazil is being referenced.
- Table 1: While I am fairly convinced armadillo hunting is an important risk factor, without a comparison group, I am not sure you can identify occupations as “risk factors”. I suggest to have a bold heading of something like, “Involvement in potentially high risk activities” rather than “Risk factors”.
- Table 1: what does fomites mean here?
- Table 1: It make be interesting to include occupations in the table, since it is included in results
- Line 197: write out CAP
- Line 231: suggest to change to “All coccidioidomycosis patients, including those involved in microepidemics, were from Northeast Brazil”.
- Line 234: what does it mean, specifically, to be “characterized by drought?” Can the authors provide some context, and a citation, about the frequency of drought? Else I suggest some rephrasing with something you can support, such as “Northeastern Brazil has a semiarid climate that is [susceptible to periods of drought (ideally a citation here)]/ [has an average annual rainfall less than XX mm], etc. ”.
- Line 238: suggest to change “probably..” to “potentially due to underreporting during the COVID-19 pandemic” in order to weaken the assumption
- Figure 3 and results: I am having trouble understanding how your models generate your results. From the brief description in Appendix, the predictors assume a linear relationship. However, in results, and in Figure 3, the authors seem to use a loess curve. Is it that the p-value relates to the linear predictor, and the statement about decreases cases beyond 100 mm precipitation is from the smoothed curve. If so, I am not fully convinced by the images that this is a strong association, and I am concerned that eyeballing the relationship cannot be used to test the hypothesis that precipitation above 100 mm is associated with fewer cases.

- Section 3: Besides p-values, the authors should state what the point estimate and 95% CI is relating environmental factors to incidence.
- Line 241: is this annual precip or precip in the seasons as stated in Appendix?
- Line 242: be more clear at first that the relationship is negative
- Line 247: suggest to avoid the word pluviometry
- Section 3: I do not believe the findings support the title of this section (see major concern).
- Section 3: as in methods, suggest to clearly separate the regression analysis from the ecological niche analysis. I also suggest a stand alone paragraph for soil testing – with additional information in methods about how this portion of the study was conducted. Also, how many of these soils were positive for *Coccidioides*?
- Line 255: I would think that the model doesn't suggest that the endemic area is larger, but that soil characteristics suitable for the fungus exists even in areas where cases have not been observed. This introduces the possibility that the fungus is there but cases have not yet been reported. Similar, line 260, rather than potential coccidioidomycosis hotspots, change to "potentially suitable habitat"

Discussion:

- Line 297: suggest to change to "where they *are thought to degrade*" to reflect the uncertainty in the field. This phrase could be supported by the genomic work of Shapton, et al. , *Genome Res*
- Line 301: "a" appears to have a cross through
- Line 303: suggest to change to "is likely underreported due to multiple.."
- Line 306: Rather than concluding that the niche modeling suggests *Coccidioides* cases are already occurring in a location, I suggest to conclude that possible habitat for the fungus exists outside the area where cases are documented (see major comment). Thus, I suggest to change to "suggest that soils and environmental conditions in other northeastern states of Brazil may be suitable for *C. posadasii* growth and sporulation."
- Line 309: This citation is for the US and I am not sure can be applied to say that cases will expand in Brazil. As an example, do warming models project that Brazil will become more arid under climate change (some areas will become wetter)? I would suggest to only use this citation if properly contextualized for Brazil.
- Line 316: might cite "think fungus" or mention CDC campaigns
- Line 318: suggest to change "cannot" to "are unlikely to"
- Line 332: suggest a word other than "miscegenation", as this word may have derogatory undertones. Perhaps mixing?
- Line 347: state how many patients had multiple lesions. Suggest to remove the word "However" from the start of this sentence.
- Line 356: I am confused by "despite being asymptomatic": the patients were not asymptomatic, were there? Is there another word that would be more appropriate here?
- Line 363: I am still working really hard to keep states and biomes straight. I suggest rewording to something easier for a reader unfamiliar with Brazil. For instance, "We found the majority of cases in Piaui, in both the Caatinga and the Cerrado biomes. As Cerrado is the dominant biome within the northeast, factors beyond biome alone must explain lower diagnoses in cases in other northeastern states."

- Line 368: citation 28 is not about Brazil and does not seem appropriate here. I also do not feel that your results support the statement that low precipitation causes “spikes” in cases.
- Line 369: cite association between dust storms and earthquakes in the US. However, I would also support removing earthquakes, as that is not a common association, and was more tied to the resultant landslide than the quake itself.
- Line 388: “is a neglected disease in marginalized populations, who, for cultural and/or economic reasons, engage in activities that expose them to infection” – from this sentence, I assume that armadillo hunting is more common among marginalized populations (probably African-American since that was your dominant cases). Is this the intent of the sentence? The authors could make this sentence more clear.
- Line 390: “We also observed that specific bioclimatic variables such as low pluviosity and extensive droughts seem key to increasing the number of cases in Brazil, a finding that resonates in the context of global warming.” See major comments. I do not believe your findings have considered “extensive droughts”. Further, is Brazil expected to experience more droughts under global warming? This is key to your point, so if so, should be cited.

Figures

Figure 2: The colors of the biome are difficult to distinguish as there are two shades of blue on the map and four shades of green. Suggest to use different colors.

Figure 3:

- A) suggest to change legend from “histogram” to “Time series”
- B and C) are these demonstrating the fit from the regression model or are they simply a loess curve?

Reviewer #3:

Remarks to the Author:

Eulalio et al. performed a large case series between 1978 (!) and 2021 to describe the epidemiological, clinical, and genomic landscape of Coccidioidomycosis in Brazil. They report a strong male:female ratio and found that armadillo hunting was the main risk factor for disease. The authors should be commended for undertaking such an endeavour as the results of their study will surely make a useful addition to the current literature. While the methods are reasonable and the data are informative, I found many aspects of the manuscript to be poorly written. I therefore suggest a thorough revision not to detract the readership from the great story the authors are sharing. Furthermore, I have several additional comments I hope the authors consider:

Major concerns:

- The inclusion criteria for study inclusion are not listed in the main manuscript.
- How were cases selected? The authors allude that patients presenting with respiratory signs were considered for inclusion but it seems incredibly unlikely that a chart review was performed for all patients with respiratory signs over a 40+ year period.
- What proportion of included patients had a matching fungal isolate? The fourth section of the results suggests that 13 isolates were recovered, which is far from a representative number. Should these results be included in this manuscript or form the basis of a second, accompanying piece? I fear that they are distracting more than they are additive in this context.
- The discussion should be shortened and a paragraph on limitations should be added.

Minor concerns:

- Why is Coccidioidomycosis abbreviated to CM? It does not shorten the word count.
- The first paragraph of the introduction can surely be shortened.
- The hypothesis in the second introductory paragraph should be omitted, as it suggests that it will be answered convincingly in the manuscript. Unfortunately, the study design precludes obtaining a definitive answer to the hypothesis.
- The third introductory paragraph is poorly written; there are many elements that belong in the methods and some could arguably be in the discussion.
-

Reviewer #1 (Remarks to the Author):

A very exciting and well-done study. This work addresses a major gap of understanding in coccidioidomycosis in South America/ Brazil. The work clearly reports risk factors, clinical presentations, and genomic epidemiology of the disease in northeastern Brazil. Relatively minor comments below.

A: Thanks for you valuable comments

Abstract:

Line 56: cases to case

A: This was modified accordingly – see line 62

Line 63: spike to increase

A: This was modified accordingly – see line 75

Lines 64 and 65: Unclear what the p values represent

A: We used a negative binomial regression model. We have included this information in the abstract and made that point clear in the Materials and Methods and in the Appendix file. See line 76

Background:

Lines 79 and 80: Indicate what Pezizomycotina and Ascomycota mean for the reader.

A: This was modified accordingly: subphylum Pezizomycotina, phylum Ascomycota. See line 94

Lines 85: Unclear what this sentence is trying to say.

A: This sentence was revised and we replace “positive sites” by “environmental reservoirs”. See line 99

Line 108: Sentence is a fragment.

A: This sentence was reformulated as follows: “The northeastern states of Piauí, Ceará, Maranhão, Pernambuco, and Bahia, known for their expansive semi-arid landscapes, have reported only a handful of cases with limited epidemiological information”. See lines 140-142

Methods:

Line 131: Rather than microepidemics, why not outbreaks?

A: Outbreaks and microepidemics are both terms used to describe the occurrence of infectious diseases, but they differ in terms of scale and scope. An outbreak typically refers to a larger and more widespread occurrence of a disease than a microepidemic. It may affect a localized community, region, or even multiple regions or countries.

Outbreaks often involve a greater number of cases and can garner significant public attention. A microepidemic, on the other hand, is a smaller-scale event. It usually occurs within a limited and well-defined population or setting, such as a specific place, a group of individuals with shared characteristics (e.g., armadillo hunters), in a localized community. Microepidemics involve a smaller number of cases compared to outbreaks.

Line 145: Missing period

A: Period added

-Would be good to describe the population seen at the Nathan Portela institute, mostly thinking about geography for patients serviced.

A: Cases were referred to either Institute of Tropical Diseases Nathan Portela or to the Pulmonology Clinic of Hospital Getúlio Vargas, both located in the Teresina city, Piauí

state. Both Institutions receives patients from the countryside and were seen by the clinical staff listed in our manuscript. See lines 185 and 186.

-Detail veterinary (dog) specifics in the clinical data.

A: We have detailed the veterinary data in the Appendix file.

-Methods do not indicate where soil acidity data was pulled from or if measured, how. Same with soil text, salinity, and other parameters.

A: We have detailed the soil collection and analysis in the M&M section 2

“Georeferencing, species niche modelling and environmental factors associated with coccidioidomycosis in Brazil” as well as in the Appendix file.

-Where the sequencing data is publicly available is not described.

A: Sequencing data, Bioproject and Biosamples were deposited at SRA/NCBI databases; see Appendix 1. We have also included a sentence in the main text mentioning the SRA accession numbers. See line 276

Results:

Line 165: Be consistent about capitalizing Northeastern

A: This was modified accordingly along the manuscript

Line 179: Consistency with types of nouns. Farmers and students; suggest saying majority were farmers (X) etc.

A: We have rephrased this sentenced in order to make more consistent as follows: “The primary occupation was farming, accounting for 167 individuals (57.2%), while students comprised the second-largest group with 57 individuals (19.5%). Trade workers represented 12 individuals (4.1%), and various other activities accounted for the remainder.” See lines 306

Line 181: Indicate years

A: This was modified accordingly

Line 219: Indicate years

A: This was modified accordingly

Discussion:

-Where is the data showing that most cases were immunocompetent? If no data, explain why it is assumed.

A: Your observation is valid. We have modified as follows: “No patients had a prior history suggestive of immunodeficiency such as recurrent infections, immune-mediated diseases, and any illness requiring immunosuppressive regimens” This was included in the Appendix file, Inclusion criteria section

-Limitations of the study need to be clearly stated including that the sample set is not derived from systematic surveillance of northeastern Brazil, thus, findings may not be representative of coccidioidomycosis in northeastern Brazil.

A: We appreciate this specific point and we do agree that this topic is missing in the discussion section. A full paragraph regarding the limitation of this study was added to discussion. See lines 613 to 639

Major gaps are an unclear description of where the data for soil characteristics comes from or if generated, how it was measured; limitations of the study of which there are key ones given the collection of samples/cases; and submission of genomic data to public repositories.

A: We agree with the suggestion and we have added an in depth description of how soils were collected and analyzed in the appendix section, a full paragraph mentioning the limitations of the study in the discussion section and we have also included a sentence in the main text/appendix sections mentioning the SRA accession numbers for genomic data.

Reviewer #2 (Remarks to the Author): Please see attached document for complete review

This paper examines coccidioidomycosis in Brazil, describing characteristics of patients, genetic variation in clinical isolates, and environmental factors associated with case and environmental detection. Coccidioidomycosis is an emerging infection and of importance to study. A strength of this article is that it is the largest case series study of the disease in South America. The authors cover a wide range of research questions. I do have some concerns with the conclusions drawn from the study, which are documented under Major comments (and some under minor comments). Additionally, I believe there is opportunity to strengthen this paper by improving clarity in key places and expanding upon methods. In the attached, I document major and minor concerns. This paper examines coccidioidomycosis in Brazil, describing characteristics of patients, genetic variation in clinical isolates, and environmental factors associated with case and environmental detection. Coccidioidomycosis is an emerging infection and of importance to study. A strength of this article is that it is the largest case series study of the disease in South America. The authors cover a wide range of research questions. I do have some concerns with the conclusions drawn from the study, which are documented under Major comments (and some under minor comments). Additionally, I believe there is opportunity to strengthen this paper by improving clarity in key places and expanding upon methods. Below I document major and minor concerns.

A: We do appreciate the valuable comments raised along the manuscript and we will respond point by point the commentaries in order to enhance the clarity of this paper, to make improvements in critical sections and elaborate on the methods, as recommended.

Major comments:

- While it may be a logical assumption to think that droughts are related to coccidioidomycosis in Brazil, I do not believe the findings of this paper can tell us about drought in Brazil for a few reasons, including 1) drought was not modelling in this analysis – only precipitation; 2) I have concerns about model specification, which are documented in the methods section. I suggest that statements about drought being key for coccidioidomycosis in Brazil be replaced with statements only about precipitation. Results should state effect estimates and 95% CI's. If the authors wish to discuss drought, they may do so, but should include citations about drought frequency, periods classified as droughts according to some definition, and climate projections for whether Brazil is expected to become drier or wetter under climate change. Alternatively, I do not believe the paper would suffer from a more descriptive approach to the time series data. For instance, cases were really high in 2015-2017 – what was happening then?

A: We acknowledge the reviewer's comment and agree that our paper primarily focuses on the relationship between precipitation and coccidioidomycosis. While drought was not explicitly modeled in this analysis, we will ensure to replace

references to drought with a focus on precipitation in the revised manuscript. To increase the robustness of the correlation between the prevalence density of coccidioidomycosis with pluviosity we performed rainfall analysis based on the annual precipitation (in mm) of the states of Maranhão and Piauí collected by 30 conventional rain gauges for the period of 1978 to 2021. Regarding the model specification: We understand your concerns about the model specification, which are mentioned in the methods section. We carefully review and address these concerns, making sure our methodology is robust and transparent and included state effect estimates and 95% CI's. Your suggestion for a more descriptive approach to the time series data is well-taken. We provided a more detailed explanation of the temporal patterns in coccidioidomycosis cases, including the high incidence observed in 2015-2017, and explore potential factors contributing to this pattern.

I worked hard to keep the states and biomes straight. I suggest re-writing several areas of this paper assuming the reader has no background on Brazil geography. I include below some areas where this could be done.

A: Your feedback on re-writing specific sections with readers unfamiliar with Brazil's geography in mind is invaluable. We have carefully reviewed your comments (especially the ones listed in the minor comments) and made the necessary revisions throughout the manuscript to enhance the overall understanding of the biomes and states in Brazil. Our goal is to ensure that the paper is as comprehensible as possible for a broader readership.

- In some areas, methods are lacking, including, as mentioned, regression model specification, as well as the soil analysis

A: Your comment regarding methodological aspects that require further attention is well-received. In the revised version, we will provide a more thorough outline of our regression model specifications, including detailed information on the variables used, model assumptions, and how these were met. We will also elaborate on the soil analysis, providing additional information on the methodologies employed, including sample collection, preparation, and analytical techniques. See M&M section 2 "Georeferencing, species niche modelling and environmental factors associated with coccidioidomycosis in Brazil" and the Appendix file.

- The authors cover so many different topics in this paper, which is both a strength and can make the paper difficult to follow. I suggest more clear delineations between research questions.
- A: To address this issue and provide readers with a more structured and focused narrative, we have implemented clearer delineations between research questions in the revised manuscript. As you suggested, we have listed the research questions explicitly, and each section of the paper now corresponds to a specific research question. This restructuring enhances the overall organization of the paper and ensures that readers can easily navigate the content while maintaining a comprehensive understanding of the various topics discussed.

comments

Abstract:

Precipitation: suggest to clarify that low precipitation is associated with incidence

A: This was clarified in the abstract. See lines 76-78

WHO's priority list [of fungal pathogens]

A: This was modified accordingly. See line 57

Line 56: change semi colon to comma

A: This was modified accordingly. See line 62

Line 64: clarify if precipitation is current year precipitation, also clarify the relationship is an inverse one

A: This was clarified accordingly. See lines 76-78

Line 66: potentially a typo - C. posadasii strains.

A: This was modified accordingly. See line 80

Line 68: suggest to use a word other than pluviosity (rainfall?) that might be more recognizable; see comments about "drought"

A: This was modified accordingly. We have made the point of drought and annual rainfall clear along the manuscript. See line 82

Background:

Line 84: would it make more sense to change animal "dens" to "burrows"

A: There this was modified accordingly throughout the text. See line 98

Line 85: The phrasing of this sentence could be edited for clarity: “The number of infections in California and Arizona ranks disease burden as one of the highest reported infectious diseases, with high morbidity and mortality”. Is this saying, “Coccidioidomycosis is one of the highest reported infectious diseases [nationally? Globally? In CA and AZ?], with most of the infectious reported from California and Arizona, and is associated with high morbidity and mortality.” Secondly, the phrase “associated with high morbidity and mortality” is vague. I might suggest to remove and include the case fatality ratio.

A: Thanks for your suggestion; this was modified to improve clarity as follows “Coccidioidomycosis is a prominent infectious disease in the western United States, primarily reported in California and Arizona, and is characterized by a noteworthy case fatality ratio”. See lines 98-101

Line 88: in addition to sources 7 and 8, more recent study has found associations between drought and incidence (Head, et al., Lancet Planetary Health).

A: Thanks for your suggestion – this reference was added to the manuscript – See line 103

Line 96: reported in Argentina?

A: Yes, the first coccidioidomycosis case was reported in Argentina – see this <https://academic.oup.com/cid/article/44/9/1202/328906>

Line 108: This sentence seems like a partial sentence, “Our unbiased and multi-institutional approach to assess genetic variation in the pathogen and environmental components that might affect the epidemiology of coccidioidomycosis”

A: We reformulated this sentence to make clear for the audience as follows “Due to the limited understanding of the disease, we embarked on an extensive evaluation of coccidioidomycosis in Brazil. Our comprehensive, unbiased, and multi-institutional approach aimed to investigate clinical profile, genetic variations within the pathogen and explore environmental factors that could influence the epidemiology of coccidioidomycosis in Brazil.” Lines 144-147

Line 111: I am a little confused about whether data is from only these two states or all of Brazil, given that results (line 168 and elsewhere) talks about other states.

A: The observations are correct – this was modified in the results section since all cases herein reported were derived from patients from Maranhão and Piauí states.

Methods:

Line 130: an outsider may not understand dogs are involved in hunting. Maybe edit “the dogs” to “of hunting dogs”

A: Thanks for your suggestion and this sentence was modified accordingly – See line 199

Line 132: how was a shared event identified?

A: Shared event of exposure were characterized by having coccidioidomycosis clinical symptoms after a same armadillo hunt or after extract stone from the same excavation/place. We have clarified the sentence as follows: “Shared exposure events were recognized through the occurrence of coccidioidomycosis clinical symptoms in individuals who had been exposed to common infectious sources (i.e. hunting an armadillo in a burrow complex or been involved in extracting stones from a common excavation site).” Lines 202-206

Clinical data: how were the 100 patients chosen?

A: The selection of the 100 patients for this study was based on the availability of comprehensive clinical records among the larger cohort of 292 patients with laboratory-proven coccidioidomycosis. We have added the following sentence to clarify this important point “These 100 patients were chosen to ensure that the study had access to the most detailed and complete clinical data.” See lines 208-209

In section 2, suggest to more clearly delineate your aims here as 1) niche modeling and 2) assess the relationship between climate variables and cases.

A: Thanks for your suggestions. We have modified the topic as follows:

Niche Modeling and Assessment of Climate Variables' Influence on "Coccidioidomycosis in Brazil". We have also modified this section as suggested. See lines 218-219

Section 2 Appendix could provide more details about the models. I also suggest at least two separate paragraphs to delineate the two different aims. If you have room. I would suggest to include some details on environmental variables in the main text.

A: We have revised the Appendix to provide more comprehensive details about the models used in our study, offering a more in-depth understanding of our methodologies. We have incorporated two separate paragraphs to clearly delineate the different aims of our study and we have included relevant details on environmental variables in the main text.

Niche modeling: what were the 19 variables obtained as predictors?

The 19 variables were obtained from the worldclim database and are the bioclimatic variables that contribute to the realized niche of a species. The 19 variables are BIO1 = Annual Mean Temperature, BIO2 = Mean Diurnal Range (Mean of monthly (max temp - min temp)), BIO3 = Isothermality (BIO2/BIO7) ($\times 100$), BIO4 = Temperature Seasonality (standard deviation $\times 100$), BIO5 = Max Temperature of Warmest Month, BIO6 = Min Temperature of Coldest Month, BIO7 = Temperature Annual Range (BIO5-BIO6), BIO8 = Mean Temperature of Wettest Quarter, BIO9 = Mean Temperature of Driest Quarter, BIO10 = Mean Temperature of Warmest Quarter, BIO11 = Mean Temperature of Coldest Quarter, BIO12 = Annual Precipitation, BIO13 = Precipitation of Wettest Month, BIO14 = Precipitation of Driest Month, BIO15 = Precipitation Seasonality (Coefficient of Variation), BIO16 = Precipitation of Wettest Quarter, BIO17 = Precipitation of Driest Quarter, BIO18 = Precipitation of Warmest Quarter, BIO19 = Precipitation of Coldest Quarter. This information is available at the Appendix 1 section (<https://www.worldclim.org/data/bioclim.html>).

Niche modeling: if two variables were autocorrelated, were both discarded? Or how did you select which ones?

A: After a test for co-linearity a "leave-one-out" Jackknife test was performed among correlated variables to assess which variable contributes best to model performance. This is made clearer in the Methods and in the Appendix as follows "Variables were assessed for collinearity with a pairwise Spearman's correlation followed by leave-one-out Jackknife test among all correlated variables in order to rule out the correlated variable that least decreased the model's performance. Model performance was assessed using the area under the curve (AUC) of the receiver operating characteristic (ROC)" See lines 228-229

Niche modeling: might be helpful to include the months of the coldest quarter and the warmest quarter. Is precipitation seasonal in Brazil? What does a term on temp seasonality mean? Also clarify that precipitation is total.

A: Those variables (coldest quarter and warmest quarter) were left out of the model because they did not contribute to the overall model performance. Temp seasonality is a bioclimatic variable and is calculated by multiplying the standard deviation by 100. The precipitation variables in the final model are defined differently. For instance, Precipitation seasonality is the coefficient of variation, however precipitation in the coldest and warmest quarters are total precipitation amounts.

Regression model: how did you aggregate case counts (e.g., over what space and time period?)

A: The coccidioidomycosis cases were aggregated from the years 1978-2020 in both Maranhao and Piaui states of northeast Brazil. This is made clearer in the manuscript in both Methods and Appendix sections. Lines 231-232

Regression model: precipitation and temperature in the current year seems troublesome as a variable because, if I am understanding correctly, some of the precipitation/temp factoring into the metric could have occurred following the case. Is it possible to examine finer temporal scales?

A: This is true however; this is the greatest resolution we can achieve because we do not when the coccidioidomycosis case was actually diagnosed at a fine scale (i.e. day of the month). Even further we cannot tell when the person was infected with the fungus because of the lag between infection and diagnosis

Regression model: precipitation over time is often correlated. Is both precipitation and precipitation one year prior put into the model? I might suggest to do so.

A: Precipitation and precipitation one-year prior (to account for lags) are both predictor variables of the final regression model. We made this clearer in the paper in both Methods and Appendix sections. Lines 235-238

Regression model: I imagine over time that population changed. How was population accounted for in your models?

A: The regression model did not take into account precipitation (i.e. cases per 100,000). However, the population of this region of Brazil has stayed fairly constant.

Regression model: what model was selected as the final model? (e.g., what were its components?)

The final model is negative-binomial regression model composed of total yearly precipitation, average yearly temperature, and total precipitation of the previous year as predictor variables. $(Y_{\text{(coccidioidomycosis cases)}} = \beta_{\text{(avg yearly temp)}} + \beta_{\text{(total yearly precipitation)}} + \beta_{\text{(total precipitation of the previous year)}} + \epsilon)$. We made this clearer in the paper, in both Methods and Appendix sections. Lines 237-238

Section 3: Include sample size for clinical isolates in main text.

A: The number of samples were included in the materials and methods section. Line 270 In results you talk about soil characteristics from microepidemics, but this is not well-described in the methods. Can the authors add more about their sampling plan (how many samples were collected and how) and how they tested for various properties (e.g., texture, salinity, pH, organics). Were these soils tested for *C. posadasii*?

A: We appreciate your suggestions and would like to provide more detail regarding our sampling plan and the analysis of soil properties. In our study, we collected nine soil samples from sites with a history of high dust exposure, such as armadillo burrows, quarries, and charcoal factories. These samples were collected aseptically in containers with screw caps to ensure an airtight seal. Approximately 1000 g of soil was collected from each site, and we took samples from at least six different points within a depth range of 10 to 30 cm below the soil surface. Regarding the analysis of soil properties, we measured key physicochemical characteristics, including texture, salinity, pH, and total organic content. The specific methods employed for these measurements were described in the Soil Analysis Methods Manual book, published by the Brazilian Agricultural Research Corporation (EMBRAPA). We conducted these analyses at the Laboratório de Análise de Solos (LASO), Federal University of Piau . However, we want to clarify that we did not test the soil samples for the presence of *C. posadasii* for this particular study. However we have already identified the fungus in armadillos DOI: 10.1023/a:1007273019647 and in burrows DOI: 10.1186/1471-2180-11-108 using molecular and microbiology techniques. This was included in the methods (lines 262-266) and Appendix section.

Results:

Line 170: the reader needs more context for caatinga and cerrado biomes. Perhaps a description could be added to introduction, or methods. Similarly, it is not clear to me what a cerrado-to- caatinga transition means. This is also true to Savana (line 233)

A: To provide more context for our readers, we have included brief descriptions of the Caatinga and Cerrado biomes in the methods section, as well as a clarification of the term 'Cerrado-to-Caatinga transition' in the results section. Additionally, we have added a concise explanation of the term 'Savana' to enhance the overall clarity of our manuscript. We hope that these additions improve the understanding of our study and address the reviewer's concerns. Lines 296-297

Identification of risk factors: it is difficult to say if these characteristics are risk factors without an understanding of the distribution of these factors in the underlying population. Is it possible to state the demographics of the underlying population? Are roughly 78% of the underlying population African American? (Is African American correct here?)

Similarly, what proportion of the underlying population are farmers? In various age groups? If this information is not known, I would just describe this as descriptive, rather than "risk factors"

A: We acknowledge the importance of understanding the distribution of these characteristics within the underlying population. While we do not have specific demographic data for the entire population in the region, we provided a more descriptive interpretation of these characteristics, recognizing that the term 'risk factors' may require a broader context. Instead of referring to these characteristics as 'risk factors,' we described them as 'demographic and clinical attributes of the patient cohort.' This terminology conveys that we are presenting the characteristics of the patients in our study without making direct claims about causality or the broader population. Regarding the ancestry: In the Brazilian context, 'African American' is not used, and we appreciate your point. We

utilized the term ' Afro-Brazilian' to describe individuals with African heritage, which is more appropriate in this context. The sentence was replaced by “In alignment with the prevalent Afro-Brazilian heritage in the northeastern region of Brazil, a majority of the patients (78.4%) shared this ancestry, while the proportion of individuals of Caucasian descent was considerably lower (20.9%, $p=5.9E-23$).” As well in various parts of the manuscript. Lines 299-301.

Line 179: “this area of Brazil” – without familiarity with Brazil, and my confusion about what states this study covers, I am not sure what part of Brazil is being referenced.

A: This was modified accordingly. See line 306

Table 1: While I am fairly convinced armadillo hunting is an important risk factor, without a comparison group, I am not sure you can identify occupations as “risk factors”. I suggest to have a bold heading of something like, “Involvement in potentially high risk activities” rather than “Risk factors”.

A: This was modified accordingly. See table 1

Table 1: what does fomites mean here?

A: This case involves the spouse of an armadillo hunter who contracted *Coccidioides* infection while cleaning pants soiled with dirt.

Table 1: It make be interesting to include occupations in the table, since it is included in results

A: The main occupations are listed in Table 1 beyond the results.

Line 197: write out CAP

A: This was modified accordingly. Line 334

Line 231: suggest to change to “All coccidioidomycosis patients, including those involved in microepidemics, were from Northeast Brazil”.

A: Thanks for your suggestions. This was modified accordingly

Line 234: what does it mean, specifically, to be “characterized by drought?” Can the authors provide some context, and a citation, about the frequency of drought? Else I suggest some rephrasing with something you can support, such as “Northeastern Brazil has a semiarid climate that is [susceptible to periods of drought (ideally a citation here)]/[has an average annual rainfall less than XX mm], etc. ”.

A: In this revised sentence, we provided additional context about the susceptibility of Northeastern Brazil to drought, explain its impact, and reference the average annual rainfall to support our statement as follows: “Northeastern Brazil is known for its susceptibility to periods of drought. The region experiences recurrent episodes of prolonged dry spells, which can have a significant impact on local climate conditions and environmental factors¹⁶” The distribution of cases” Lines 376-378

Line 238: suggest to change “probably..” to “potentially due to underreporting during the COVID- 19 pandemic” in order to weaken the assumption

A: This was modified accordingly. See line 382

Figure 3 and results: I am having trouble understanding how your models generate your results. From the brief description in Appendix, the predictors assume a linear relationship. However, in results, and in Figure 3, the authors seem to use a loess curve.

A: We apologize for not making this clearer, the model we ran to gather these results was a negative binomial regression model not a linear regression. Therefore, in order to generate the fit line, we utilized the loess smoother function based on the negative binomial distribution not a normal distribution. We mad this clearer in the manuscript.

Is it that the p-value relates to the linear predictor, and the statement about decreases cases beyond 100 mm precipitation is from the smoothed curve. If so, I am not fully convinced by the images that this is a strong association, and I am concerned that eyeballing the relationship cannot be used to test the hypothesis that precipitation above 100 mm is associated with fewer cases.

A: The p-values are calculated from the results from the negative-binomial regression model described above and the best of fit loess smooth curve is also based on the negative binomial distribution. We agree that the language about “precipitation above 100mm is associated with fewer cases” is too strong as we did not test this directly. We remove this statement and indicated that “There appears to be a negative relationship with high precipitation leading to fewer coccidioidomycosis cases deficient rainfall leads to severe drought impacts” Lines 389-391

Section 3: Besides p-values, the authors should state what the point estimate and 95% CI is relating environmental factors to incidence.

A: Thank you for this suggestion we add the CI for these variables. Lines 386-389

Line 241: is this annual precip or precip in the seasons as stated in Appendix?

A: This is total yearly precipitation and total yearly precipitation of one year prior to case as stated in the appendix

Line 242: be more clear at first that the relationship is negative

A: Thank you - we made this more clear in the manuscript. Line 392

Line 247: suggest to avoid the word pluviometry

A: We will change this to total precipitation. Line 393-419

Section 3: I do not believe the findings support the title of this section (see major concern).

A: We agree, as it stands, we did not measure drought, we will change language as well as the section title.

Section 3: as in methods, suggest to clearly separate the regression analysis from the ecological niche analysis. I also suggest a stand alone paragraph for soil testing – with additional information in methods about how this portion of the study was conducted. Also, how many of these soils were positive for *Coccidioides*?

A: We have restructured this section to clearly separate the regression analysis and the ecological niche analysis. We have introduced distinct sections for these analyses to enhance the organization and readability of the paper. We have included a dedicated paragraph in the Methods and Appendix sections to provide more detailed information on how the soil testing was conducted. Additionally, we have included information about the number of soil samples that tested positive for *Coccidioides*. In addition a more suitable title for section could be: "3 – The Role of Low Annual Precipitation in *Coccidioidomycosis* Prevalence in Northeastern Brazil." This title clearly conveys the section's content and the focus of the analysis. By implementing these changes, we aim to improve the clarity and organization of the manuscript, addressing the reviewer's valuable feedback. We have also included a stand-alone paragraph for soil testing. We haven't tested for the presence of *Coccidioides* in those soils in this particular manuscript. However the fungus was already identified in those areas in armadillos DOI: 10.1023/a:1007273019647 and in burrows DOI: 10.1186/1471-2180-11-108 using molecular and microbiology techniques.

Line 255: I would think that the model doesn't suggest that the endemic area is larger, but that soil characteristics suitable for the fungus exists even in areas where cases have not been observed. This introduces the possibility that the fungus is there but cases have not yet been reported. Similar, line 260, rather than potential coccidioidomycosis hotspots, change to “potentially suitable habitat”

A: To address the reviewer's concerns and provide a more accurate description of the findings, we revised the two statements as follows: "Species niche modeling suggests that soil characteristics suitable for the fungus exist in areas where coccidioidomycosis cases have not yet been reported, indicating the potential for a larger endemic area than previously observed." and "However, the model identifies areas in Northern Bahia, Sergipe, Alagoas, Pernambuco, Paraiba, and Rio Grande do Norte states as potentially suitable habitats for coccidioidomycosis, suggesting the presence of soil conditions conducive to the fungus, even in regions where cases have not been documented." These

revisions clarify that the model is identifying suitable habitat conditions for the fungus rather than definitively stating the presence of the disease or cases, aligning with the reviewer's feedback. 420-429

Discussion:

Line 297: suggest to change to “where they are thought to degrade” to reflect the uncertainty in the field. This phrase could be supported by the genomic work of Shapton, et al. , Genome Res

A: This was modified accordingly and the reference was added. Line 496

Line 301: “a” appears to have a cross through

A: This was corrected accordingly. Line 498

Line 303: suggest to change to “is likely underreported due to multiple..”

This was modified accordingly. Line 505

Line 306: Rather than concluding that the niche modeling suggests Coccidioides cases are already occurring in a location, I suggest to conclude that possible habitat for the fungus exists outside the area where cases are documented (see major comment). Thus, I suggest to change to “suggest that soils and environmental conditions in other northeastern states of Brazil may be suitable for C. posadasii growth and sporulation.”

A: Thanks for your suggestion and this was modified accordingly. See line 511-513

Line 309: This citation is for the US and I am not sure can be applied to say that cases will expand in Brazil. As an example, do warming models project that Brazil will become more arid under climate change (some areas will become wetter)? I would suggest to only use this citation if properly contextualized for Brazil.

A: We have included a study that identify areas in Northeast Brazil that will become more arid under climate change. Line 514

Line 316: might cite “think fungus” or mention CDC campaigns

A: We have added the following sentence “Initiatives like the CDC's "Fungal Disease Awareness Week" campaign are instrumental in cultivating this mycological paradigm shift among professionals” Line 519-520

Line 318: suggest to change “cannot” to “are unlikely to”

A: This was modified accordingly. Line 522

Line 332: suggest a word other than “miscegenation”, as this word may have derogatory undertones. Perhaps mixing?

A: Thanks for your suggestion – Indeed “intermixing” is more proper in here – Line 547

Line 347: state how many patients had multiple lesions. Suggest to remove the word “However” from the start of this sentence.

A: This was modified accordingly – Line 563

Line 356: I am confused by “despite being asymptomatic”: the patients were not asymptomatic, were there? Is there another word that would be more appropriate here?

A: We have revised the sentence as follows: “Although clinically symptomatic and having an otherwise normal physical examination, multiple nodules were observed, either calcified or not, which differs from other studies in which residual lesions were lower (2%-39.61%).” Line 572-574

Line 363: I am still working really hard to keep states and biomes straight. I suggest rewording to something easier for a reader unfamiliar with Brazil. For instance, “We found the majority of cases in Piauí, in both the Caatinga and the Cerrado biomes. As Cerrado is the dominant biome within the northeast, factors beyond biome alone must explain lower diagnoses in cases in other northeastern states.”

A: We have modified the sentence to improve clarity as follows: “We found the majority of cases in Piauí, in both the Caatinga and the Cerrado biomes. As Caatinga is the dominant biome within the northeast, factors beyond biome alone must explain lower

diagnoses in cases in other northeastern states.” We would like to mention to the reviewer that the Caatinga is the dominant Biome of Northeast Brazil. See line 589-591

Line 368: citation 28 is not about Brazil and does not seem appropriate here. I also do not feel that your results support the statement that low precipitation causes “spikes” in cases.

A: We have modified the sentence as follows: “The increase in cases in Brazil was associated with low annual precipitation, as indicated by the bioclimatic variables of Precipitation and Precipitation one-year prior. It’s worth noting that a similar trend has been observed in the United States³²”. Also, instead of “spikes” we used “increase”.
Line 595-598

Line 369: cite association between dust storms and earthquakes in the US. However, I would also support removing earthquakes, as that is not a common association, and was more tied to the resultant landslide than the quake itself.

A: We agree and this sentence was modified accordingly. Instead of earthquakes we replaced by earthquake-induced landslides – Line 598

Line 388: “is a neglected disease in marginalized populations, who, for cultural and/or economic reasons, engage in activities that expose them to infection” – from this sentence, I assume that armadillo hunting is more common among marginalized populations (probably African-American since that was your dominant cases). Is this the intent of the sentence? The authors could make this sentence more clear.

A: Yes, the intent of the sentence is to suggest that armadillo hunting, which is a known risk factor for coccidioidomycosis, is more common among marginalized populations in Brazil. The reference to North America having a more diverse population getting infected implies that the mode of infection and the demographics of affected individuals differ between the two regions. To make this sentence clearer, we rephrase it as follows: “In Brazil, coccidioidomycosis predominantly impacts marginalized populations, specifically individuals involved in activities like armadillo hunting. This practice holds a unique environmental, economic, and cultural importance, particularly for the subsistence of rural communities in this context. Contrary to North America, where infection risk extends to a diverse demographic, these findings underscore disparities in infection mechanisms and the makeup of affected populations in the two regions.” Lines 644-650

Line 390: “We also observed that specific bioclimatic variables such as low pluviosity and extensive droughts seem key to increasing the number of cases in Brazil, a finding that resonates in the context of global warming.” See major comments. I do not believe your findings have considered “extensive droughts”. Further, is Brazil expected to experience more droughts under global warming? This is key to your point, so if so, should be cited.

A: We appreciate the comments. We have performed additional analysis that indicated that high endemic areas of the disease in Brazil has experienced severe droughts compared to areas that has lower incidence of the disease. Therefore we have modified the text accordingly as well highlighted that Northeast Brazil is expected to experience severe droughts under global warming: “Our observations indicate that particular bioclimatic variables, such as reduced precipitation, play a pivotal role in the escalation

of cases in Brazil. These findings are particularly relevant within the framework of global climate change in Northeastern portion of Brazil²¹ Lines 650-653

Figures

Figure 2: The colors of the biome are difficult to distinguish as there are two shades of blue on the map and four shades of green. Suggest to use different colors.

A: We have changed the color scheme to improve clarity of biome limits

Figure 3:

A) suggest to change legend from “histogram” to “Time series”

B and C) are these demonstrating the fit from the regression model or are they simply a loess curve?

A: “histogram” was replaced by “Time series”. Yes loess curves were shown

Reviewer #3 (Remarks to the Author):

Eulalio et al. performed a large case series between 1978 (!) and 2021 to describe the epidemiological, clinical, and genomic landscape of *Coccidioidomycosis* in Brazil. They report a strong male:female ratio and found that armadillo hunting was the main risk factor for disease. The authors should be commended for undertaking such an endeavour as the results of their study will surely make a useful addition to the current literature. While the methods are reasonable and the data are informative, I found many aspects of the manuscript to be poorly written. I therefore suggest a thorough revision not to detract the readership from the great story the authors are sharing. Furthermore, I have several additional comments I hope the authors consider:

A: We sincerely appreciate the reviewer's valuable comments and feedback. We are committed to addressing the issues raised in the review comprehensively and ensuring that the manuscript is revised to meet the highest standards of clarity and scientific rigor. Thank you for your constructive input, and we look forward to enhancing the quality of the manuscript based on your suggestions.

Major concerns:

- The inclusion criteria for study inclusion are not listed in the main manuscript.

A: We have included a more detailed “inclusion criteria” into the main text. We have also expanded this section into the Appendix document.

- How were cases selected? The authors allude that patients presenting with respiratory signs were considered for inclusion but it seems incredibly unlikely that a chart review was performed for all patients with respiratory signs over a 40+ year period.

A: Thanks for your comments. We included a sentence clarifying this – Lines 208-209

- What proportion of included patients had a matching fungal isolate? The fourth section of the results suggests that 13 isolates were recovered, which is far from a representative number. Should these results be included in this manuscript or form the basis of a second, accompanying piece? I fear that they are distracting more than they are additive in this context.

A: We appreciate the comments. We included the information about the 13 isolates to provide some insights into the genetic diversity and relatedness of the *Coccidioides* strains in the region. While this number might seem limited, it's important to note that this data spans a decade, and it also complements the previously genotyped isolates. We weren't planning to conduct extensive genome-wide association studies in this specific manuscript, but rather to investigate the genotypes contributing to the disease in Brazil. Understanding the genomic backgrounds of pathogens is crucial for managing outbreaks, conducting evolutionary analyses, and enhancing the molecular epidemiology of the pathogen, especially in the context of One Health, where the interaction between human, animal, and environmental health is considered. The information is meant to contribute to a broader understanding of the disease dynamics and its genetic aspects.

- The discussion should be shortened and a paragraph on limitations should be added.

A: Thanks for your suggestion. We have shortened the discussion as possible as well as included a paragraph on limitations of the study

Minor concerns:

- Why is Coccidioidomycosis abbreviated to CM? It does not shorten the word count.

A: You are correct – we have removed the abbreviation “CM” that only appeared in the first sentence of the background section

- The first paragraph of the introduction can surely be shortened.

A: We have shortened the first paragraph as possible

- The hypothesis in the second introductory paragraph should be omitted, as it suggests that it will be answered convincingly in the manuscript. Unfortunately, the study design precludes obtaining a definitive answer to the hypothesis.

A: Thanks for your suggestions. We smother the statements as we haven't test all those hypothesis. We have modified the sentence as follows: “We posit that the lower incidence in Latin America compared to North America may be attributed to factors such as altered virulence traits, underdiagnosis and underreporting, and variations in bioclimatic and demographic conditions” Lines 134-137

- The third introductory paragraph is poorly written; there are many elements that belong in the methods and some could arguably be in the discussion.

A: We reformulated the third introductory paragraph as suggested. Those aspects were also added to the discussion section. “Due to the limited understanding of the disease, we embarked on an extensive evaluation of coccidioidomycosis in Brazil. Our comprehensive, unbiased, and multi-institutional approach aimed to investigate clinical profile, genetic variations within the pathogen and explore environmental factors that could influence the epidemiology of coccidioidomycosis in Brazil.” Lines 143-147

Reviewers' Comments:

Reviewer #2:

Remarks to the Author:

I thank the authors for their revised version, which I do find much improved. I have remaining questions about the regression model specification and interpretation. I document these in the attached report.

I thank the authors for their attention to detail in this revision. I believe that the manuscript is clearer and comes to conclusions that are better supported by the data. The authors have presented a wealth of data and analyses, for which they should be commended. I particularly appreciate the addition of some descriptive elements.

I appreciate the author's revisions with respect to the regression model and its interpretation. However, I still have a few concerns/questions regarding the specification of the regression model and results. I have included my concerns below for the authors to concern. However, it might be possible that the data do not support robust regression modeling, and the authors might consider simply reporting some descriptive results about trends in cases and trends in precipitation over the study period.

- The authors geocode the cases to municipality, but then use the spatial aggregation of state level for regression modeling. Why is this done – is the data too sparse at the municipal level or the environmental predictors not available? It seems it would be better to do a regression model at the municipal level (including an offset on population) if it is possible (although I understand it might not be). Similarly, the spatial aggregation for the niche modeling is unclear to me. The appendix suggests they are aggregated to state level for niche modeling too but this was not my understanding from the main text.
- Relatedly, in the response, the authors stated, “The coccidioidomycosis cases were aggregated from the years 1978-2020 in both Maranhao and Piaui states of northeast Brazil. This is made clearer in the manuscript in both Methods and Appendix sections. Lines 231-232”. I do not see this in the methods in these lines and still feel that this point needs to be made clearer. I would also suggest to edit that text to specify clearly that the aggregation is year and state: “Coccidioidomycosis cases were aggregated yearly and at the state level between 1978-2020 in both Maranhao and Piaui states of northeast Brazil”
- Is the model implemented separately for each state? Line 295 implies so, but this is not clear in line 177. If this is not the case, then I believe that the authors should adjust for state in their model to account for repeated observations over time.
- Line 177: Having the model written out is helpful for understanding the terms, but I am concerned the model is not totally accurate. For instance – this is a negative binomial model, so did the authors use a log link function? This should be reflected in the model. It would be helpful to see subscripts for t in the model and for i (if both states are included).
- Interpretation of results: are the results only given for the state of Piaui? Is this because that state had more cases and is the only state for which a regression model was built, or are the authors only reporting a state for which a relationship existed? If the latter, it may be helpful to say so. If the authors want to model both states, why not include them both in the same model, and adjust for state?
- Interpretation of results, line 297: If I am understanding correctly, the authors use a negative binomial model, which has a log link function. If this is true, the interpretation in this sentence is not correct: “The point estimates calculated by the model for precipitation are -0.01 (95% CI Interval -0.02 to -0.002) and for precipitation one year prior is -0.01 (95% CI Interval -0.02 to -0.005). In other words, for every one unit increase in precipitation the number of coccidioidomycosis cases is expected to decrease by 0.01.” The proper interpretation is to exponentiate the model coefficients and interpret the effect as a ratio measure. For instance,

$\exp(-0.01) = 0.99$. Then you would interpret it as, for every one unit increase in precipitation the number of coccidioidomycosis cases is expected to decrease by 1% (1-0.99)

I additionally have a few outstanding minor comments:

- Line 99: reported where? Argentina or all of South/Central America?
- Line 113 – 118: this sentence is a bit confusingly worded: “used geographical coordinates related to clinical, veterinary cases, and environmental detection coupled with bioclimatic and demographic factors to predict the true geographic range of the disease in Brazil”
- Line 138: I suggest to include a transition sentence as to why you care about hunting dogs. For instance, “Hunting armadillos was identified as a common activity prior to infection. Accordingly, for cases with reported exposure to dust from armadillo habitats, we also followed up and confirmed the infection status of any dog involved in the hunting.”
- Line 148: I appreciate the revision, but feel it isn’t entirely clear. I suggest something like, “These 100 patients were chosen on the basis of having complete medical records.”
- Line 188: Suggest an opening sentence to this paragraph to motivate what you are doing. For instance, “Lastly, we assessed characteristics of the soil from locations of known exposures.”
- Line 196: Suggest an opening sentence to show your objective (e.g., In order to understand X, we did Y).
- Line 221: What is meant by “particularly” in the latter half of this sentence: “Reported cases span both caatinga and cerrado biomes, *particularly the semi-arid cerrado-to-caatinga transition, a zone where both biomes meet and intermingle.*” Does this mean that most cases fell in the transition zone?
- Line 280: could “finding” be changed to “abnormalities”?
- Line 329: how do the authors categorize “low”, “medium”, and “high” organic matter?
- Figure 2: I appreciate the author’s efforts to improve clarity of this map. I remain a little confused by the two shades of blue/green/purple. I think that the darker shade is just delineating the two states where cases occurred. Perhaps this can be indicated in the legend?
- Appendix, “The coccidioidomycosis cases were aggregated from the years 1978-2020 in both Maranhao and Piaui states of Northeast Brazil for species niche modelling”. Were cases aggregated to state for the niche modeling? This feels like a very large aggregation when you have finer spatial resolution on the cases. Note also some typos with periods in this section.

Reviewer #3:
Remarks to the Author:
None

I thank the authors for their attention to detail in this revision. I believe that the manuscript is clearer and comes to conclusions that are better supported by the data. The authors have presented a wealth of data and analyses, for which they should be commended. I particularly appreciate the addition of some descriptive elements.

I appreciate the author's revisions with respect to the regression model and its interpretation. However, I still have a few concerns/questions regarding the specification of the regression model and results. I have included my concerns below for the authors to concern. However, it might be possible that the data do not support robust regression modeling, and the authors might consider simply reporting some descriptive results about trends in cases and trends in precipitation over the study period.

The authors geocode the cases to municipality, but then use the spatial aggregation of state level for regression modeling. Why is this done – is the data too sparse at the municipal level or the environmental predictors not available? It seems it would be better to do a regression model at the municipal level (including an offset on population) if it is possible (although I understand it might not be). Similarly, the spatial aggregation for the niche modeling is unclear to me. The appendix suggests they are aggregated to state level for niche modeling too but this was not my understanding from the main text.

A: We greatly appreciate your thorough review and insightful feedback on our manuscript. Your comments have been instrumental in refining our work, and we're grateful for your attention to detail. Regarding the specification of the regression model and the spatial aggregation for niche modeling, we'd like to provide clarification on the rationale behind our approach. The decision to aggregate cases to the state level for modeling was primarily due to the limitations in the resolution of available precipitation and temperature data. Unfortunately, we encountered constraints in obtaining finer-resolution environmental predictors necessary for municipal-level analysis. As such, the state level served as the highest resolution achievable with the available data for this specific analysis. We completely agree that conducting the analysis at a higher resolution would provide a more detailed understanding. Regrettably, due to the unavailability of requisite environmental predictors at the municipal level, performing the regression model with finer granularity was not feasible within the scope of this study. In the case of niche modeling, it seems there might have been some confusion, and we apologize for any lack of clarity in the manuscript. The cases utilized for niche modeling were indeed geo-referenced using longitude and latitude coordinates, providing a higher resolution than the state level. We will ensure that this aspect is elucidated more explicitly in the main text to avoid any ambiguity.

Relatedly, in the response, the authors stated, "The coccidioidomycosis cases were aggregated from the years 1978-2020 in both Maranhao and Piaui states of northeast Brazil. This is made clearer in the manuscript in both Methods and Appendix sections. Lines 231-232". I do not see this in the methods in these lines and still feel that this point needs to be made clearer. I would also suggest to edit that text to specify clearly that the aggregation is year and state: "Coccidioidomycosis cases were aggregated yearly and at the state level between 1978-2020 in both Maranhao and Piaui states of northeast Brazil"

A: Thank you for your insightful observations regarding the clarification of our methodology concerning the aggregation of coccidioidomycosis cases in Maranhao and Piaui states of northeast Brazil. To address your point, we have updated the Methods section to provide a clearer description of the temporal and spatial aggregation of the coccidioidomycosis cases. The revised text now explicitly states: "Coccidioidomycosis cases were aggregated yearly and at the state level between 1978-2020 in both Maranhao and Piaui states of northeast Brazil." – See lines 221-223

Is the model implemented separately for each state? Line 295 implies so, but this is not clear in line 177. If this is not the case, then I believe that the authors should adjust for state in their model to account for repeated observations over time.

A: The best fitting model includes each state as a predictor variable in the model. There are not separate models for each state. We have made this clearer and more fixed the equation.

Line 177: Having the model written out is helpful for understanding the terms, but I am concerned the model is not totally accurate. For instance – this is a negative binomial model, so did the authors use a log link function? This should be reflected in the model. It would be helpful to see subscripts for t in the model and for i (if both states are included).

A: We have incorporated the log link function into the model, addressing this aspect for greater accuracy and transparency. Additionally, the state beta term has been included in the model equation to reflect its significance in our analysis as follows:

$$Y(\text{Vf cases}) = \beta(\text{avg yearly temp}) + \beta(\text{total yearly precipitation}) + \beta(\text{total precipitation of the previous year}) + \beta(\text{State}) + \epsilon$$

This is a simplified version of the equation for negative binomial regression in order for the readers to visualize the model and each predictor, we feel adding all the coefficients in, such as t in the simple equation would over complicate. Our aim was to strike a balance between accuracy and readability to ensure that the model's essence is conveyed without unnecessary complexity.

Below is the detailed model:

Negative binomial regression model is based on the Poisson-gamma mixed distribution which is useful for predicting count-based data such as disease case counts across time and space. This is defined by :

$$P(Y_i = y_i) = \frac{\mu_i^{y_i} \exp(-\mu_i)}{y_i!}$$

Where P(.) shows the probability of Y *Coccidioides* infected people observed in the *i*th state (Piaui and Marahoa) from 1978-2020. *Y_i* are count values (0,1,2,3...) and μ_i represents the expected frequency of *Coccidioides* cases as a function of the explanatory variables *x_i* where:

$$\ln(\mu_i) = x_i^T \beta$$

Where β represents the vector of explanatory variables (climate variables and Brazilian state). The vector coefficients are estimated by maximizing the logarithm of the likelihood function:

$$\ln L(\beta) = \sum_i [-\exp(x_i^T \beta) + (x_i^T \beta)y_i - \ln y_i !]$$

The following model was then put into effect:

$$\begin{aligned} \ln(\text{Number of Cases of Coccidioidomycosis}) \\ = \beta_o + \beta_{(\text{avg yearly temperature})} + \beta_{(\text{total yealry precipitation})} \\ + \beta_{(\text{total precipitation of the previous year})} + \beta_{(\text{Brazillian State})} + \varepsilon_i \end{aligned}$$

Where ε_i is the gamma distributed error term.

Equations are adapted from Oztig and Askin 2020. We hope this amended approach better aligns with both accuracy and readability standards.

Interpretation of results: are the results only given for the state of Piauí? Is this because that state had more cases and is the only state for which a regression model was built, or are the authors only reporting a state for which a relationship existed? If the latter, it may be helpful to say so. If the authors want to model both states, why not include them both in the same model, and adjust for state?

A: The regression model conducted revealed that the state of Piauí significantly predicts Valley fever cases ($p < 0.0001$), while the state of Maranhão does not demonstrate significant predictive value ($p = 0.5$). Our analysis includes data from both states; however, we report results for Piauí due to the significant relationship identified within that region. The expected log case count in Piauí is 1.35 times greater than that of Maranhão. Notably, the effects of precipitation and temperature variables encompass data from both states, indicating their comprehensive consideration despite the focus on reporting significant relationships found solely in Piauí. We have included the following sentence to clarify this point as follows: “We are only reporting the results of Piauí state because there are significant relationships to precipitation and temperature” – See lines 350-351

Interpretation of results, line 297: If I am understanding correctly, the authors use a negative binomial model, which has a log link function. If this is true, the interpretation in this sentence is not correct: “The point estimates calculated by the model for precipitation are -0.01 (95% CI Interval -0.02 to -0.002) and for precipitation one year prior is -0.01 (95% CI Interval -0.02 to 0.005). In other words, for every one unit increase in precipitation the number of coccidioidomycosis cases is expected to decrease by 0.01.” The proper interpretation is to exponentiate the model coefficients and interpret the effect as a ratio measure. For instance, $\exp(-0.01) = 0.99$. Then you would interpret it as, for every one unit increase in precipitation the number of coccidioidomycosis cases is expected to decrease by 1% ($1 - 0.99$)

A: Thank you for catching this. We thought it would be easier to interpret by reporting the un-exponentiated coefficients but we have changed this as follows: “For this state, precipitation (Fig. 3B, $p=0.015$) and precipitation one-year prior (Fig. 3C, $p=0.001$) were significant predictors. The point estimates calculated by the model for precipitation are $\exp(-0.01)=0.99$ (95% CI Interval $0.98 \leftrightarrow 0.99$) and for precipitation one year prior is $\exp(-0.01)=0.99$ (95% CI Interval $0.98 \leftrightarrow 0.99$). In other words, for every one unit increase in precipitation there is a the number of Vf cases is expected to decrease by 1%. There appears to be a negative relationship between high precipitation and coccidioidomycosis.” – See lines 353-359

I additionally have a few outstanding minor comments:

Line 99: reported where? Argentina or all of South/Central America?

A: Thanks for catching this. We have replaced this sentence by “Interestingly, 100 years after the first report of the disease in Argentina, fewer than 1,000 total cases have been reported in Central and South America” – See lines 130-131

Line 113 – 118: this sentence is a bit confusingly worded: “used geographical coordinates related to clinical, veterinary cases, and environmental detection coupled with bioclimatic and demographic factors to predict the true geographic range of the disease in Brazil”

A: We have improved this sentence as follows: “integrated geocoded clinical and veterinary cases along with environmental detections of *Coccidioides*, amalgamating these data sets with bioclimatic and demographic factors. This comprehensive approach allowed us to accurately forecast the geographic extent of the disease in Brazil” – See lines 148-151

Line 138: I suggest to include a transition sentence as to why you care about hunting dogs. For instance, “Hunting armadillos was identified as a common activity prior to infection. Accordingly, for cases with reported exposure to dust from armadillo habitats, we also followed up and confirmed the infection status of any dog involved in the hunting.”

A: We appreciate your suggestion to include a transitional sentence regarding the significance of hunting dogs in the context of our study. The suggestions were incorporated into the manuscript. See lines 177-179

Line 148: I appreciate the revision, but feel it isn’t entirely clear. I suggest something like, “These 100 patients were chosen on the basis of having complete medical records.”

A: We appreciate your suggestion, and this was incorporated into the manuscript. See lines 188 - 189

Line 188: Suggest an opening sentence to this paragraph to motivate what you are doing. For instance, “Lastly, we assessed characteristics of the soil from locations of known exposures.”

A: We appreciate your suggestion, and this was modified accordingly. See lines 238-239

Line 196: Suggest an opening sentence to show your objective (e.g., In order to understand X, we did Y).

A: Thanks for your suggestion and this was incorporated into the manuscript as follows: “To unravel the genetic diversity and population structure of *Coccidioides posadasii* in Brazil, our study conducted fungal isolation, whole-genome sequencing, and evolutionary comparisons with additional *Coccidioides* sp. genomes.” See lines 246-248

Line 221: What is meant by “particularly” in the latter half of this sentence: “Reported cases span both caatinga and cerrado biomes, particularly the semi-arid cerrado-to-caatinga transition, a zone where both biomes meet and intermingle.” Does this mean that most cases fell in the transition zone?

A: No, we would like to state that the disease is found in both biomes, including the cerrado-to-caatinga transition. Thus, we replaced “particularly” by “including”. See line 277.

Line 280: could “finding” be changed to “abnormalities”?

A: This was replaced accordingly. See line 337

Line 329: how do the authors categorize “low”, “medium”, and “high” organic matter?

A: This classification was based on the USDA (United States Department of Agriculture) as follows: Low Organic Matter Content: Typically, soil with organic carbon content below 1-2% is considered to have low organic matter content. Medium Organic Matter Content: Soil with organic carbon content between 2-3.5% might be categorized as having medium organic matter content. High Organic Matter Content: Soil with organic carbon content above 3.5-6% or higher can be considered to have high organic matter content. This was included into the appendix section “Physicochemical characteristics of the soil samples”

Figure 2: I appreciate the author’s efforts to improve clarity of this map. I remain a little confused by the two shades of blue/green/purple. I think that the darker shade is just delineating the two states where cases occurred. Perhaps this can be indicated in the legend?

A: This was indicated in the figure legend as suggested. See lines 587-588

Appendix, “The coccidioidomycosis cases were aggregated from the years 1978-2020 in both Maranhao and Piaui states of Northeast Brazil for species niche modelling”. Were cases aggregated to state for the niche modeling? This feels like a very large aggregation when you have finer spatial resolution on the cases. Note also some typos with periods in this section.

A: No, the cases were not aggregated by state for niche modeling this was a typo and is corrected. See Appendix1

Reviewers' Comments:

Reviewer #2:

Remarks to the Author:

I thank the authors for their revision, which has improved clarity. Questions remain about the interpretation of the model results. Namely, I do not understand how the authors arrive at associations between precipitation and cases for one state only, when they used a model that included both states. The authors clearly state in the revision, "There are not separate models for each state".

Additionally, I believe that the authors should continue to make edits to ensure that the model specification and coefficient interpretation are properly described in the text. I offer suggestions for how to do this in the attached

Model presentation in the methods.

First, I understand the desire for interpretability, but presented models must be accurate. If the authors feel that the correct model specification is too detailed for the main text, they should clearly describe the model accurately in words, and then they can include the correct model specification in the supplement. I suggest the following rewrite of this paragraph to:

In the Northeastern Brazilian states of Maranhao and Piauí, we investigated the relationship between climate and coccidioidomycosis cases from 1978 to 2020 using negative-binomial regression models. Coccidioidomycosis cases were aggregated yearly and at the state level between 1978-2020 in both Maranhao and Piauí states of Northeast Brazil for regression modeling. The model included a fixed effects for total yearly precipitation, average yearly temperature, total precipitation from the previous year, and state. The final model is presented in the Supplement. The model was screened for autocorrelation among predictor variables, ensuring that all retained variables maintained Pearson's correlation coefficients below 0.6, enhancing the model's validity. Moreover Rainfall data were collected from the public repository of the Instituto Nacional de Meteorologia (<https://portal.inmet.gov.br/>). Daily rainfall data of the States of Maranhão 187 and Piauí were collected by 30 conventional rain gauges for the period of 1978 to 2021. Annual precipitation (in mm) was obtained by summing the daily data by rain gauge. Mean annual precipitation was calculated by station. Data interpolation was made based on the analysis of variograms and ordinary kriging for the average of the period. Calculations and modeling were performed with the R packages 'sf' 15, 'spatstat' 16, 'stars' 17, and 'MASS' (cite).

In blue, I highlighted the phrase, Mean annual precipitation was calculated by station. Is this needed as the authors are using total precipitation?

In the model supplement, I suggest writing the following as the model:

$$\ln(Y_{it}) = \beta_0 + \beta_1(\text{avg. yearly temp}_{it}) + \beta_2(\text{total precip in year } t_{it}) + \beta_3(\text{total precip in year } t - 1_{it-1}) + \beta_4(\text{state}_i) + \varepsilon$$

Where Y_{it} is the number of coccidioidomycosis cases in state i and year t

This specification is both accurate yet simple.

Presentation of model results

Second, the results continue to state that the relationship between precipitation and Valley fever exists only for Piauí. The authors write:

“We are only reporting the results of Piauí state because there are significant relationships to precipitation and temperature”.

It remains unclear to me how a significant relationship between precipitation can exist only for one state based on the model specification. Given the model includes state as a fixed effect, there should only be one coefficient for total yearly precipitation in the model, and it cannot be interpreted as being state-specific. The authors do not describe conducting stratified models nor models with an interaction term between precipitation and state.

Furthermore, in the revision justification, the authors state:

The regression model conducted revealed that the state of Piauí significantly predicts Valley fever cases ($p < 0.0001$), while the state of Maranhão does not demonstrate significant predictive value ($p = 0.5$).

I do not understand how this statement can be obtained from the model. The model includes state as a fixed effect, where the reference value appears to be Maranhão. Thus, there should only be one p-value on state. Exponentiation of the coefficient yields the relative cases Piauí compared to Maranhão (the reference).

That issue aside, I believe that the presentation of model results could be cleaner than what the authors have proposed. I suggest a rewrite of the following text starting line 301:

Distribution of cases across time was uneven: in 2004 and between 2015 and 2017 there was a spike of cases, particularly in the state of Piauí (Fig. 3A). There were no significant associations between precipitation and temperature detected for Maranhão state. In Piauí state, we detected a significant inverse association between coccidioidomycosis cases and precipitation in the current year (Fig. 3B, $p = 0.015$) and one-year prior (Fig. 3C, $p = 0.001$). The point estimates calculated by the model for precipitation in the current and in the prior year were both -0.01 . Exponentiating those coefficients, we find that, holding temperature and precipitation in the prior year constant, every millimeter increase in precipitation in the current year, is associated with 1% fewer coccidioidomycosis cases (IRR: 0.99, 95% CI: 0.98, 0.99). Similarly, holding temperature and precipitation in the current year constant, for every millimeter increase in precipitation in the prior year is associated with 1% fewer coccidioidomycosis cases (IRR: 0.99, 95% CI: 0.98, 0.99).

As a side note, the statement “The expected log case count in Piauí is 1.35 times greater than that of Maranhão.” in the revision is an incorrect interpretation of a model coefficient from a negative binomial model. Exponentiating coefficients yields *relative differences* of case counts. Not exponentiating coefficients yields *absolute differences* in *log* case counts. If you exponentiated the coefficient, the interpretation of the effect on state should be: “The expected log case count in Piauí is 1.35 times greater than that of Maranhão”. If you did not exponentiate it, the interpretation would be: “The expected log case count in Piauí is 1.35 higher than the log case count in Maranhão. The authors should double check that they fully understand the interpretation of model coefficients from a negative binomial model.

Model presentation in the methods.

First, I understand the desire for interpretability, but presented models must be accurate. If the authors feel that the correct model specification is too detailed for the main text, they should clearly describe the model accurately in words, and then they can include the correct model specification in the supplement. I suggest the following rewrite of this paragraph to:

In the Northeastern Brazilian states of Maranhao and Piauí, we investigated the relationship between climate and coccidioidomycosis cases from 1978 to 2020 using negative-binomial regression models. Coccidioidomycosis cases were aggregated yearly and at the state level between 1978-2020 in both Maranhao and Piauí states of Northeast Brazil for regression modeling. The model included a fixed effects for total yearly precipitation, average yearly temperature, total precipitation from the previous year, and state. The final model is presented in the Supplement. The model was screened for autocorrelation among predictor variables, ensuring that all retained variables maintained Pearson's correlation coefficients below 0.6, enhancing the model's validity. Moreover Rainfall data were collected from the public repository of the Instituto Nacional de Meteorologia (<https://portal.inmet.gov.br/>). Daily rainfall data of the States of Maranhão 187 and Piauí were collected by 30 conventional rain gauges for the period of 1978 to 2021. Annual precipitation (in mm) was obtained by summing the daily data by rain gauge. Mean annual precipitation was calculated by station. Data interpolation was made based on the analysis of variograms and ordinary kriging for the average of the period. Calculations and modeling were performed with the R packages 'sf' 15, 'spatstat' 16, 'stars' 17, and 'MASS' (cite).

In blue, I highlighted the phrase, Mean annual precipitation was calculated by station. Is this needed as the authors are using total precipitation?

A: We have made all alterations accordingly. See lines 217-282

In the model supplement, I suggest writing the following as the model:

$$\ln(Y_{it}) = \beta_0 + \beta_1 (\text{avg. yearly temp}_{it}) + \beta_2 (\text{total precip}_{in\ year\ t\ it}) + \beta_3 (\text{total precip}_{in\ year\ t-1\ it-1}) + \beta_4 (\text{state } i) + \varepsilon_{it}$$

Where Y_{it} is the number of coccidioidomycosis cases in state i and year t

This specification is both accurate yet simple.

A: We have modified the equation and moved to Appendix section as suggested by the reviewer. See Appendix 1, "Coccidioides niche modeling and relationship between climate and coccidioidomycosis cases in Brazil"

Presentation of model results

Second, the results continue to state that the relationship between precipitation and Valley fever exists only for Piauí. The authors write:

“We are only reporting the results of Piauí state because there are significant relationships to precipitation and temperature”.

It remains unclear to me how a significant relationship between precipitation can exist only for one state based on the model specification. Given the model includes state as a fixed effect, there should only be one coefficient for total yearly precipitation in the model, and it cannot be interpreted as being statespecific. The authors do not describe conducting stratified models nor models with an interaction term between precipitation and state.

A: We removed this sentence to avoid misinterpretation of the results.

Furthermore, in the revision justification, the authors state: The regression model conducted revealed that the state of Piauí significantly predicts Valley fever cases ($p < 0.0001$), while the state of Maranhão does not demonstrate significant predictive value ($p = 0.5$).

I do not understand how this statement can be obtained from the model. The model includes state as a fixed effect, where the reference value appears to be Maranhão. Thus, there should only be one p-value on state. Exponentiation of the coefficient yields the relative cases Piauí compared to Maranhão (the reference).

A: That was a mistake in the response to reviewer letter. We apologize. The corrected sentence is: “The regression model conducted revealed that the state of Piauí significantly predicts Valley fever cases ($p < 0.0001$) when compared to Maranhão.” I hope this makes things cleared out.

That issue aside, I believe that the presentation of model results could be cleaner than what the authors have proposed. I suggest a rewrite of the following text starting line 301:

Distribution of cases across time was uneven: in 2004 and between 2015 and 2017 there was a spike of cases, particularly in the state of Piauí (Fig. 3A). There were no significant associations between precipitation and temperature detected for Maranhão state. In Piauí state, we detected a significant inverse association between coccidioidomycosis cases and precipitation in the current year (Fig. 3B, $p = 0.015$) and one-year prior (Fig. 3C, $p = 0.001$). The point estimates calculated by the model for precipitation in the current and in the prior year were both -0.01 . Exponentiating those coefficients, we find that, holding temperature and precipitation in the prior year constant, every millimeter increase in precipitation in the current year, is associated with 1% fewer coccidioidomycosis cases (IRR: 0.99, 95% CI: 0.98, 0.99). Similarly, holding

temperature and precipitation in the current year constant, for every millimeter increase in precipitation in the prior year is associated with 1% fewer coccidioidomycosis cases (IRR: 0.99, 95% CI: 0.98, 0.99).

A: We rephrased these sentences according the reviewer's suggestions. See lines 342-357.

As a side note, the statement "The expected log case count in Piauí is 1.35 times greater than that of Maranhão." in the revision is an incorrect interpretation of a model coefficient from a negative binomial model. Exponentiating coefficients yields relative differences of case counts. Not exponentiating coefficients yields absolute differences in log case counts. If you exponentiated the coefficient, the interpretation of the effect on state should be: "The expected log case count in Piauí is 1.35 times greater than that of Maranhão". If you did not exponentiate it, the interpretation would be: "The expected log case count in Piauí is 1.35 higher than the log case count in Maranhão. The authors should double check that they fully understand the interpretation of model coefficients from a negative binomial model.

A: Yes you are correct. The expected log case count in Piauí is 1.35 higher than the log case count in Maranhão. Again, this was an error of omission in the last response to the reviewer's letter. I hope this makes things cleared out.

Reviewers' Comments:

Reviewer #2:

Remarks to the Author:

The authors have addressed my concerns. I have a few remaining suggestions:

Most importantly, the authors should verify that this statement is indeed true, as my assumption was that the relationship was assessed while controlling for state, but not assessed for each state:

"There were no significant associations between precipitation and temperature detected for Maranhao state. In Piaui state, we detected a significant inverse association between coccidioidomycosis cases and precipitation in the current year (Fig. 3B, $p=0.015$) and one-year prior (Fig. 3C, $p=0.001$)."

From my understanding, the association was across both states, in which state this statement should not imply that the relationship with precipitation varies by state. It could say,

"We detected a significant inverse association between coccidioidomycosis cases and precipitation in the current year (Fig. 3B, $p=0.015$) and one-year prior (Fig. 3C, $p=0.001$)"

A few additional comments include:

Results Line 287: In the statement, "The role of low annual precipitation in coccidioidomycosis prevalence in Northeastern Brazil", suggest to change prevalence to incidence

Results Line 309: Remove word "for" from "for every millimeter" to make improve grammar

Discussion, line 450: don't need to capitalize Precipitation

Discussion line 493: suggest to say "may play a pivotal role." rather than "play a pivotal role."

Response to the authors

Reviewer #2 (Remarks to the Author):

The authors have addressed my concerns. I have a few remaining suggestions:

Most importantly, the authors should verify that this statement is indeed true, as my assumption was that the relationship was assessed while controlling for state, but not assessed for each state:

"There were no significant associations between precipitation and temperature detected for Maranhao state. In Piaui state, we detected a significant inverse association between coccidioidomycosis cases and precipitation in the current year (Fig. 3B, $p=0.015$) and one-year prior (Fig. 3C, $p=0.001$)."

From my understanding, the association was across both states, in which state this statement should not imply that the relationship with precipitation varies by state. It could say,

"We detected a significant inverse association between coccidioidomycosis cases and precipitation in the current year (Fig. 3B, $p=0.015$) and one-year prior (Fig. 3C, $p=0.001$)"

A: Thanks for the suggestions. To avoid misinterpretation we have followed the reviewer's suggestions. See lines 209-211

A few additional comments include:

Results Line 287: In the statement, "The role of low annual precipitation in coccidioidomycosis prevalence in Northeastern Brazil", suggest to change prevalence to incidence

A: We have modified the text accordingly. See line 196.

Results Line 309: Remove word "for" from "for every millimeter" to make improve grammar

A: We have removed the work "for" in the refereed sentence. See line 217.

Discussion, line 450: don't need to capitalize Precipitation

A: We have modified the text accordingly. See line 357.

Discussion line 493: suggest to say "may play a pivotal role." rather than "play a pivotal role."

A: We agree with the suggestion and modified text accordingly. See line 400.